# Novel sound exposure drives dynamic changes in auditory lateralization that are associated with perceptual learning in zebra finches

Basilio Furest Cataldo [1][✉], Lillian Yang[2], Bryan Cabezas[1], Jonathan Ovetsky[1] & David S. Vicario [1][✉]

Songbirds provide a model for adult plasticity in the auditory cortex as a function of recent experience due to parallels with human auditory processing. As for speech processing in humans, activity in songbirds' higher auditory cortex (*caudomedial nidopallium*, NCM) is lateralized for complex vocalization sounds. However, in Zebra finches exposed to a novel heterospecific (canary) acoustic environment for 4–9 days, the typical pattern of right-lateralization is reversed. We now report that, in birds passively exposed to a novel heterospecific environment for extended periods (up to 21 days), the right-lateralized pattern of epidural auditory potentials first reverses transiently then returns to the typical pattern. Using acute, bilateral multi-unit electrophysiology, we confirm that this dynamic pattern occurs in NCM. Furthermore, extended exposure enhances discrimination for heterospecific stimuli. We conclude that lateralization is functionally labile and, when engaged by novel sensory experience, contributes to discrimination of novel stimuli that may be ethologically relevant. Future studies seek to determine whether, (1) the dynamicity of lateralized processes engaged by novel sensory experiences recurs with every novel challenge in the same organism; (2) the dynamic pattern extends to other cortical, thalamic or midbrain structures; and (3) the phenomenon generalizes across sensory modalities.

[1] Rutgers University, Department of Psychology, Piscataway, NJ 08854, USA. [2] The City College of New York (CUNY), Physiology, Pharmacology and Neuroscience Department, New York, NY 10031, USA. [✉]email: bf287@scarletmail.rutgers.edu; vicario@psych.rutgers.edu

Perceptual filters, either hardwired or formed early in development, provide an initial means of parsing the incoming sensory stream; they combine individual receptive fields to play important roles in auditory selectivity for complex sounds. However, these filters may not be fixed, and may be updated by subsequent auditory input such that, even in an adult organism, the auditory system undergoes plastic changes to achieve a more efficient representation of the recent statistics of the auditory environment. This fine-tuning of perceptual filters to match sounds in the current acoustic space may manifest as improved discriminability of auditory stimuli that may be ethologically relevant. For example, auditory neural plasticity in the form of tonotopic map expansion and improved behavioral performance after training, can be seen in adult animals tasked to learn stimulus-consequence associations[1,2]. Human studies of hearing loss[3], animal lesion studies[4], and studies of exposure to target acoustic stimuli during operant conditioning[5,6] all indicate that responses in the auditory cortex are modifiable in adulthood, even by passive exposure to altered statistics of the external acoustic environment[2,7–10]. Given that this plasticity is retained in adulthood, the distribution of sounds in the current environment can update perceptual filters that may in turn be maintained by the distribution of current inputs and depend on ongoing/continued exposure to sounds.

Indeed, anyone who has attempted to learn a second language in adulthood is aware that plasticity in auditory processing can persist beyond a childhood critical period. Over the course of immersion in a foreign language[11], what initially sounds like a stream of gibberish eventually separates itself into distinct, though still incomprehensible, syllables and sentences[12,13]. Although neural mechanisms underlying plasticity in auditory cortical maps and receptive fields have been identified in studies that used simple sounds[1,5,14], it is unclear whether these same mechanisms contribute to the immediate and long term changes in auditory perception seen after exposure to complex stimuli with novel acoustic parameters (e.g. a foreign language[15]), or the vocalizations of another species[16].

The songbird provides the best-developed animal model for the study of speech acquisition and perception of communication signals. Young songbirds, like human infants, learn their songs from an adult tutor model during an early critical period[17]. Initial patterning of songbird auditory responses may be genetically determined or established by early auditory experience, leading to a response bias for conspecific vocalizations similar to the acquired selective response to phonemes in the native language in humans[17,18]. In addition, motor production and sensory perception of complex vocalizations are lateralized in songbirds, as they are in humans[19–24], and lateralized sensory processing has been demonstrated in other mammals, e.g. bats[25] and monkeys[26]. Lastly, recent studies have suggested close analogies, if not homologies, between the thalamocortical auditory system of birds and mammals[27–32]

Although juvenile male Zebra finches (ZFs) learn to produce their song during the first post-hatching year, sensory learning and memory for song is not restricted to the critical period for song acquisition. Male songs are used not only for reproductive purposes, i.e., mate attraction, but also in a social context for individual recognition[33]. The ability to remember, and properly discriminate, between a neighbor's song and that of an intruder has important ethological impacts on survival[34,35]. The ability to form new auditory memories and adapt to a changing auditory landscape may be subserved by residual plasticity in adulthood that is unlocked when the auditory cortex encounters novel stimuli.

One songbird auditory brain region that exhibits plasticity in adult ZFs is the *caudomedial nidopallium* (NCM). NCM is analogous to superficial layers of mammalian A1 or perhaps a higher-order auditory cortical region[30,36]. NCM neurons respond selectively to species-specific vocalizations and undergo stimulus-specific adaptation to the unique songs of individual conspecifics, a form of long-lasting neuronal memory[37–40]. Lateralization of auditory processing in NCM has been demonstrated in electrophysiological, immediate early gene, and neurogenesis studies[24,40,41]. Neural responses to song are typically higher and adapt faster in right NCM, although the patterns of lateralization differ according to the experimental methodology and dependent measures[42]. In electrophysiological studies, where stronger responses are recorded in the right hemisphere of adults, this typical right-lateralization requires normal exposure to conspecific song during the juvenile period[24]. However, typical asymmetry can be reversed by exposure to a novel class of acoustic stimuli in adulthood, perhaps in response to altered stimulus statistics[16]. This reversal is seen after 4 and 9 days of exposure to novel heterospecific sounds, but it is unknown whether the reversal persists with longer exposure and if it has any implications for behavior.

Using a chronic sampling technique, based on epidural recordings[43], that assessed auditory responses bilaterally within the same subjects over weeks, we document the full time course of shifts in lateralization, both during a silent baseline period (thought to serve as a control condition of stability in the acoustic environment) and as a function of continuous exposure to novel acoustic environments (the experimental condition of a changing acoustic environment). We confirm the reversal reported previously[16], and go on to show that, after ~14 days (14d) of exposure to the heterospecific environment (i.e. a novel and ethologically foreign acoustic environment), lateralization returns to the typical pattern (i.e. higher right hemisphere activity). This dynamic pattern of reversal and return is confirmed by depth microelectrode recordings from NCM at critical time-points over a 30-day period. Furthermore, we show significant improvements in behavioral discrimination for heterospecific sounds only in a group of birds that received 14d or more of heterospecific exposure, and thus presumably had undergone both reversal and return of the typical lateralization pattern. Unexpectedly, recordings during the silent condition showed no lateral differences in either epidural or NCM responses, suggesting that continuous exposure to complex sounds is needed for the typical lateralized patterns to be maintained over time. The results suggest that perceptual filters in NCM remain labile in adulthood and that transient shifts in lateralized NCM activity may contribute to and/or reflect enhanced auditory discrimination for ethologically-relevant sounds.

## Results

**Time course of transient shifts in lateralization.** To assess the effect of sustained exposure to a novel acoustic environment on lateralization, adult male Zebra finches (ZFs) were exposed to either a novel heterospecific (canary aviary recording; Het-Env) or a novel conspecific (foreign Zebra finch aviary recording; Con-Env) acoustic environment (see Fig. 1). During these exposures, electrophysiological epidural activity was recorded from individual ZFs longitudinally, at regular intervals over a period of 21 days. Bilateral brain responses to a set of novel, simpler vocalizations (conspecific female ZF long-calls, 200–300 ms in duration) were assessed in each subject by recording event-related potentials (ERPs) via epidural electrodes. This enabled changes in the patterns of activity in the two hemispheres to be measured chronically. The results, computed as the Lateralization Index (LI; see Methods), show dynamic changes as a function of time induced by Het-Env exposure, as seen in a representative subject

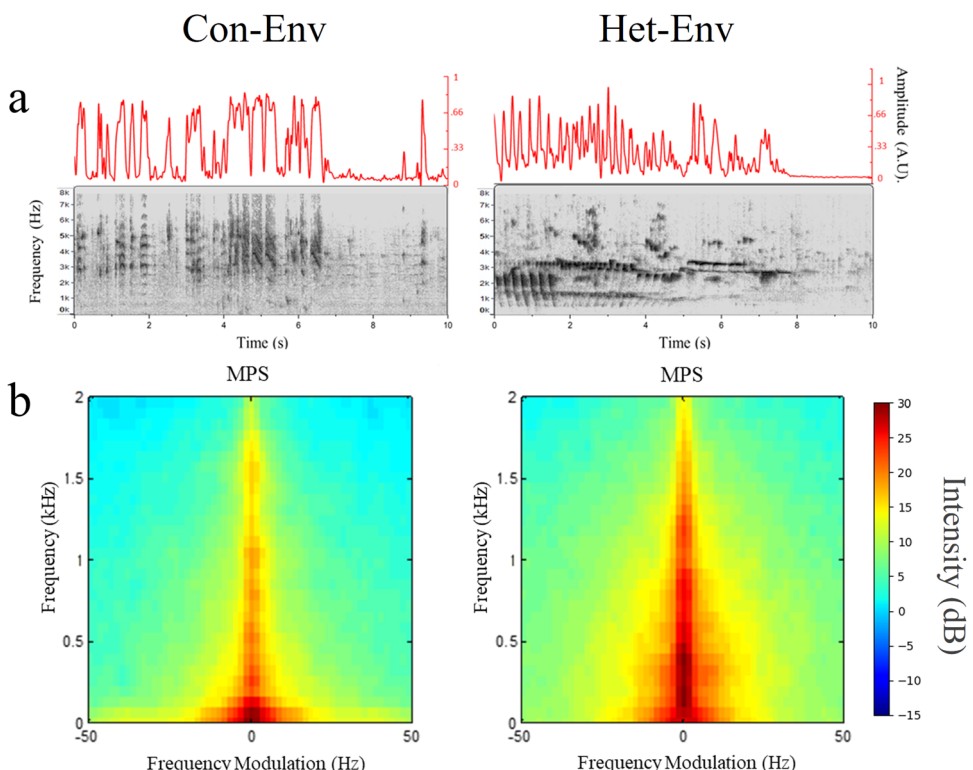

**Fig. 1 Visual and quantitative comparison between zebra finch (Con-Env) and canary (Het-Env) acoustic environments.** Visual depictions and quantifications of the Con-Env and Het-Env acoustic environments. **a** Sonograms (bottom plots) and smoothed positive amplitude envelopes (top plots) of Con-Env and Het-Env from a representative ~10 s sample of each environment. The smoothed amplitude envelope was produced by first finding extreme values of the sound signal, interpolating them via a cubic spline, and performing adjacent smoothing (averaged points were set to 500; OriginPro, 2020b). **b** Modulation Power Spectra (MPS) generated from Zebra finch (Con-Env) and canary (Het-Env) playback tapes show the temporal and spectral modulations of vocalizations recorded in the two species environments. The X axis represents temporal modulations of power over the sampling period. The Y axis represents the spectral modulation of power over short intervals of time across the frequency spectrum. Het-Env stimuli contain relatively more energy at higher frequencies and longer durations than Con-Env stimuli. MPSs were adapted from Fig. 2a in Yang and Vicario, 2015, Neuroscience, Copyright Elsevier[16].

(Fig. 2a). Auditory response lateralization reverses transiently from a typical right-side bias (LI > 0) to a left-side bias (LI < 0), then returns to the initial right-side bias (LI > 0). Group data (Fig. 2b) shows a similar dynamic pattern in birds exposed to Het-Env (red line in Fig. 2b; $n = 4$) but not in birds exposed to a conspecific acoustic environment (blue line in Fig. 2b; Con-Env; $n = 4$). There was a significant effect of acoustic environment on the LI ($F_{(1, 45)} = 5.78$, $p = 0.02$), however time of exposure did not significantly affect LI ($F_{(7, 45)} = 1.79$, $p = 0.11$). Contrary to expectation based on previous observations, the interaction between environment and time was not significant (likely due to the non-linearity of the reversal and return, and individual differences in the time required to induce shifts in lateralized activity), although there was a trend towards significance $(F_{(7, 45)} = 2.096$, $p = 0.06)$. To explore whether the environment effect was related to the duration of exposure, *post-hoc* comparisons between groups were carried out for all time bins. Of the tested comparisons, there were only significant differences between Con-Env and Het-Env at time bins 6–8 days, 9–11 days, and 12–14 days ($t_{(45)} = -3.620$, $p = 0.0004$; $t_{(45)} = -3.755$, $p = 0.0002$; $t_{(45)} = -3.419$, $p = 0.0007$, respectively); comparisons across the remaining time bins (i.e. 0–2 days, 3–5 days, 15–17 days, 18–20 days) were not significant. (Fig. 2b). Overall, the results show that exposure to Het-Env induces dynamic changes in the bias of lateralization different from those seen with Con-Env: the emergence of atypical left-biased responses; the post-hoc analyses suggests that these changes occur during a

specific period of exposure, followed by a return to the typical pattern. In addition, there was a discrepancy whereby, during the first few days (at time bin 0–2), the exemplar Het-Env individual (Fig. 2a) displayed a qualitatively left-biased activity (i.e. LI < 0) and the group-averaged Het-Env data (Fig. 2b) showed no pattern of lateralized activity (i.e. LI = 0); in addition, there were no significant differences between Het-Env and Con-Env at the 0-2d time bin. While there may be individual differences in the timepoints at which the shifts in lateralization take place, the current experiment employed a silence baseline period which was not used in previous experiments[16]. This discrepancy is addressed in the next experiment.

**Effect of silence on lateralization.** Earlier results using acute microelectrode recordings of multi-unit activity (MUA) in NCM typically showed right-biased activity[16,24]. However, the present ERP data showed little or no lateral difference at baseline or in the first 1–2 days after playback begins (Fig. 2b). This raised the question of whether this discrepancy is due to the different recording methods (ERP vs. MUA) or in experimental conditions. In the present ERP paradigm, birds were removed from their aviary and spent 4 days in silent isolation prior to the start of playback. In contrast, the earlier MUA recordings were made either in birds taken directly from a large Zebra finch aviary[24], or in birds that had already spent several days listening to conspecific acoustic playbacks[16] before being recorded. To address

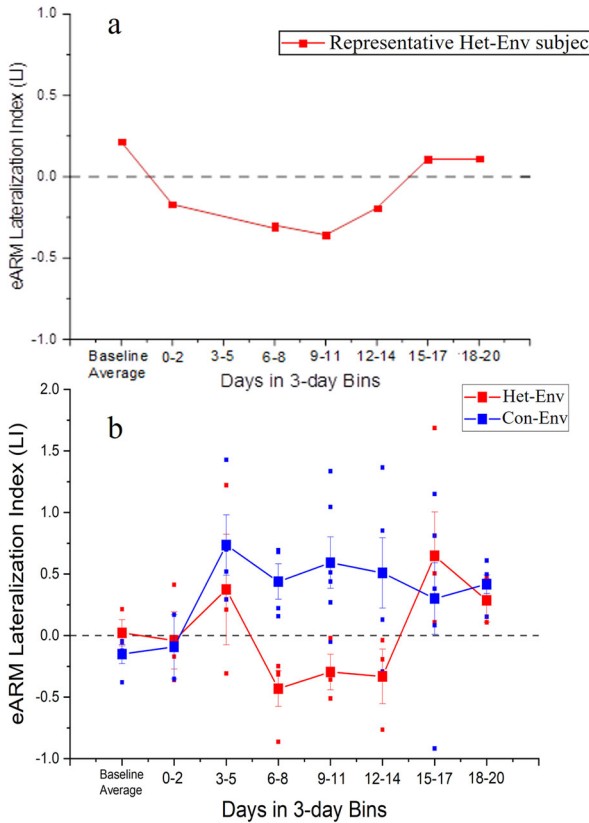

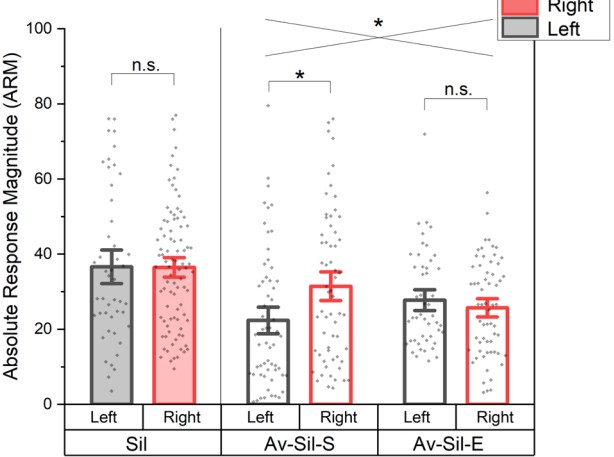

**Fig. 3 Effect of silence on lateralized NCM activity.** Comparison of lateralized NCM response magnitudes in cohorts subjected to silent conditions. Zebra finches recorded after 5 days for silence (Sil) displayed not lateralization bias. Birds recorded directly from the aviary (Av-Sil-S) display expected, typically right-biased patterns of activity. Lateralized patterns of NCM activation were not present in cohorts moved from the aviary to the silence condition (Av-Sil-E). In the comparison between the Av-Sil group, there was an interaction between hemisphere and condition, suggesting that lateralized patterns of activity depended on environment exposure. Error lines denote +− 1.5 s.e.m., (*) denotes significance at the .05 alpha level, intersecting lines denote interaction, and n.s. denotes non-significance of comparison.

**Fig. 2 Changes in lateralized epidural activity as a function of time and acoustic environment.** Timecourse of changes in lateralized epidural activity as characterized by eARMs LI. **a** Example timecourse from a single Het-Env-exposed bird. **b** Effect of exposure environment (Het-Env or Con-Env) on eARM LIs as a function of days of exposure (herein as 3-day bins). Baseline LIs were similar between both groups and suggested attenuated hemisphere-biased activity. On average, LIs were significantly different between Het-Env and Con-Env groups at three distinct time-bins: 6–8, 9–11, 12–14. Birds exposed to Het-Env displayed left-biased (<0) LIs around days 6–14 and right-biased (>0) LIs around days 15–20. In the case of figure (**b**), large square symbols reflect group-averaged eARM LIs for each time bin and error bars denote +/− 1.5. s.e.m; colored dots denote individual data points based on acoustic environment. No changes in LIs were observed in birds exposed to Con-Env. (*) denotes significant differences at the 0.05 alpha level.

the role of acoustic isolation, two separate cohorts of birds ($n = 9$) were placed in silent isolation for 5 days. Multi-unit activity (MUA) was then assessed via bilateral multiple microelectrode recordings in NCM either at the end of the 5d isolation period in one group (Sil; $n = 5$) or at both the start (Av-Sil-S; $n = 4$) and end (Av-Sil-E; same birds as Av-Sil-S) of the isolation period in the other group. MUA for each recording site was then used to calculate absolute response magnitudes (ARMs; *see Methods*) such that response strengths could be compared between hemispheres.

As seen in Fig. 3, recordings from birds housed in silence for 5 days (Sil) showed no lateral differences in NCM activity ($t(154) = -0.39$, $p = 0.69$). Data from the chronic-acute cohort (Av-Sil time-points: Av-Sil-S and Av-Sil-E) was loaded onto a factorial ANOVA, with hemisphere (H) and time-point (T) as factors, including their interaction term (HxT). Both the H and T factors did not load significantly ($F(1, 227) = 2.62$, $p = 0.11$ and $F(1, 227) = 0.008$, $p = 0.93$, respectively). However, as seen in Fig. 3, the interaction between these factors (i.e. HxT) was

significant ($F(1, 227) = 6.54$, $p = 0.01$). This significance appears to be mainly driven by the higher right-lateralized activity in the Av-Sil-S time-point ($t(227) = 3.04$, $p = 0.01$), which conforms with previous observations that, when continuously exposed to conspecifics, Zebra finches display right-lateralized NCM activity[16]. In summary, birds recorded straight from the aviary (Av-Sil-S) displayed right-lateralized activity in NCM, as expected from previous work. Additionally, ZFs, whose lateralized activity was retested following 5 days of acoustic isolation (i.e. Av-Sil-E), or ZFs that were tested only after acoustic isolation (i.e. Sil), displayed no significant lateralized NCM activity. Together, both methodological approaches yielded the same pattern of results: lateralized activity in NCM is attenuated following a period of silence. This suggests that lateralization may be maintained by, and require, ongoing exposure to complex acoustic signals.

**Dynamic lateralization in higher auditory area NCM.** To examine one localized source of the dynamic shifts in lateralized activity that were observed with epidural recordings, acute microelectrode recordings were carried out in NCM, in a separate cohort of ZFs. In these experiments, MUA was recorded from multiple microelectrodes placed bilaterally in NCM in birds that had been exposed to either Het-Env or Con-Env for various periods of time (4, 9, 14 or 30 days) corresponding to the temporal pattern of changes observed in the first experiment (Fig. 2b). Auditory responses were quantified in two ways: as absolute response magnitudes and as adaptation rates (ARMs and AdRs, respectively; *see Methods*). Data collected at 4d and 9d have been previously published[17] and are included here as part of a larger data set for comparison. As described in that earlier publication[16], control birds exposed to Con-Env for 4d or 9d exhibited the normal pattern of lateralization: higher ARMs and faster AdRs in NCM on the right side (Fig. 4a, top-left panel). In contrast, birds exposed to Het-Env for 4d or 9d had a reversed

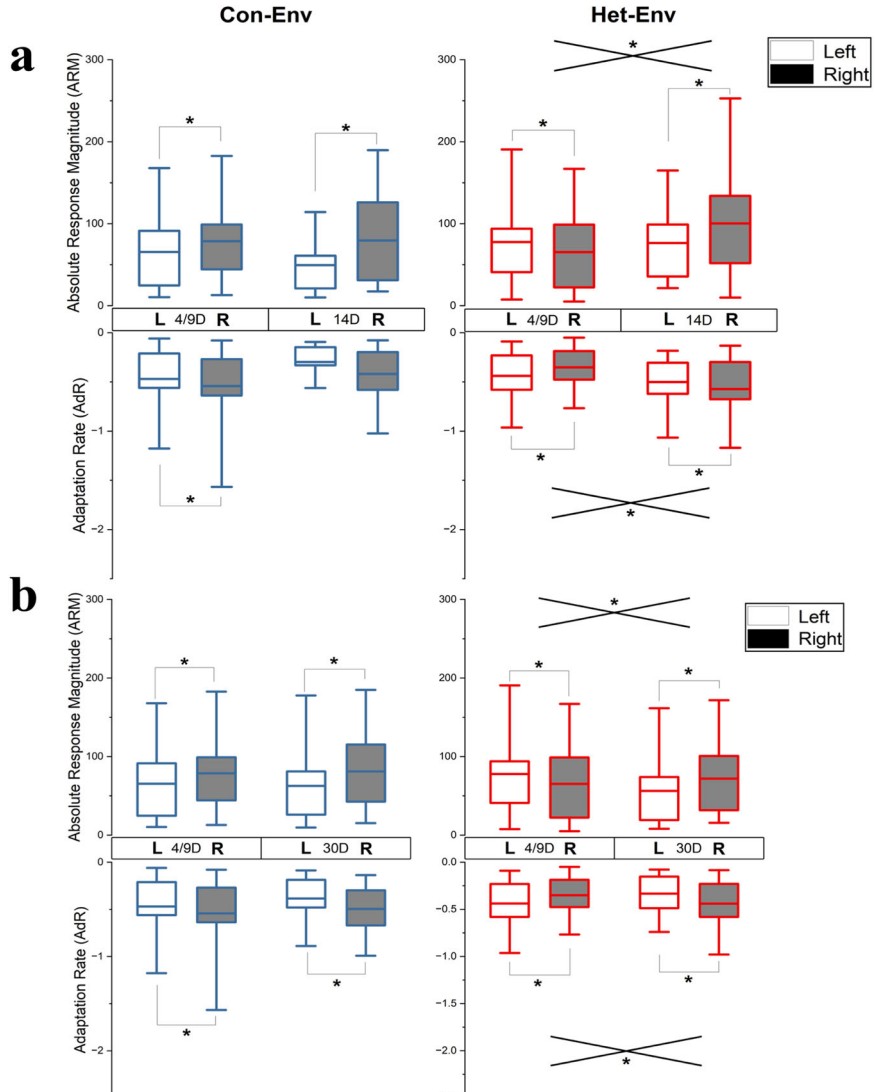

**Fig. 4 Effect of acoustic experience and duration of exposure on lateralized NCM activity.** Effects of novel environmental exposure, for different time periods, on Absolute Response Magnitudes (ARMs) and Adaptation Rates (AdRs). Illustrated, are comparison between left and right NCM ARMs and Adaptation Rates in relation to exposure type (Con-Env [blue] and Het-Env [red]) and duration (4/9D, 14D, and 30D). **a** Exposure to Het-Env for 4 and 9 days (4/9D, data collapsed) yielded significantly higher ARMs and adaptation rates in the left hemisphere, a pattern that was reversed in a group exposed to the same environment for 14 days (14D). Such changes were not observed in cohorts exposed to Con-Env for the same time periods. **b** Exposure to Het-Env for 4 and 9 days (4/9D, data collapsed) yielded significantly higher ARMs and adaptation rates in the left hemisphere, a pattern that was reversed in a group exposed to the same environment for 30 days (30D). Such changes were not observed in cohorts exposed to Con-Env for the same time periods. The data from the 4/9D exposure groups has been previously published[16] and is used for comparison against the current data (14D and 30D). The boxplots illustrate the interquartile range (box; 25–75% of the data), the whiskers denote the 5–95% data-range, and the horizontal lines denote the distributions' means. (*) denotes significance at the 0.05 alpha level, and intersecting lines denote significant interaction between hemisphere and exposure-time factors. Data for the 4/9 timepoints has been previously published[16] and is here used for comparison purposes.

pattern of lateralization with higher ARMs and faster AdRs in the left hemisphere (Fig. 4a, top-right panel).

In order to further evaluate the pattern of dynamic changes in lateralization, the present study extends this timeline by presenting additional MUA data from microelectrode recordings in NCM made after longer exposures (14 and 30 days) to the two acoustic environments. After 14d exposure, ARMs in Het-Env were larger on the right side and not different from ARMs in Con-Env. There was no main effect of housing environment when 14d Het-Env ARMs were compared to 14d Con-Env ARMs in a repeated measures ANCOVA with housing environment and hemisphere as factors. However, there was a significant main effect of hemisphere ($F(1, 199) = 10.036$, $p = 0.00178$, Fig. 4a, top

panels): the right hemisphere had higher ARMs than the left in both 14d Con-Env and 14d Het-Env groups. Furthermore, there was a significant effect of stimulus type ($F(1, 199) = 13.569$, $p = 0.0003$), indicating a strong response bias for zebra finch stimuli in both groups. Interestingly, there was also an interaction between hemisphere and stimulus type ($F(1, 199) = 6.392$, $p = 0.01224$) due to a stronger response bias for zebra finch stimulus in the right hemisphere; in fact, ARMs to zebra finch and canary stimuli were approximately equal in the left hemisphere. There is a trend towards a three-way interaction between environment, hemisphere and stimulus type ($F(1, 199) = 2.9992$, $p = 0.0849$, n.s.) due to a high conspecific bias in both hemispheres in the 14d Con-Env group; however only the right

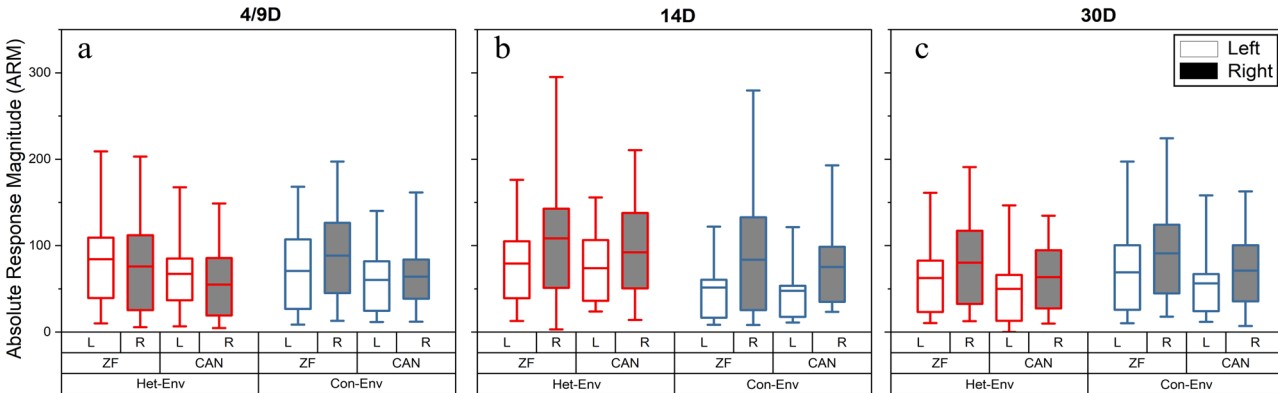

**Fig. 5 Effect of acoustic experience and duration of exposure on lateralized NCM activity as a function of test stimulus (canary vs zebra finch).** Effects of novel environmental exposure, for different time periods, on Absolute Response Magnitudes (ARMs) broken out by test stimulus. Illustrated, are the effects of auditory environment (Con-Env blue; Het-Env red) as a function of 4/9D (**a**), 14D (**b**), or 30D (**c**) exposure duration on lateralized multi-unit absolute response magnitudes (ARMs) does not depend on test stimulus (canary or zebra finch). All of the observed changes in lateralized ARMs, particularly in ZFs exposed to Het-Env for 4/9D (**a**) wherein higher left-lateralized ARMs are observed relative to any other group, do not depend on the test stimulus. For example, in panel (**a**), Het-Env ZFs exhibit higher activity in the left hemisphere for both zebra finch and canary test stimuli. For the other groups, where right-lateralized activity is observed, both zebra finch and canary stimuli drive right-lateralized activity. The data from the 4/9D exposure groups has been previously published[16] and is used for comparison against the current data (14D and 30D). The boxplots illustrate the interquartile range (box; 25–75% of the data), the whiskers denote the 5–95% data-range, and the horizontal lines denote the distributions' means.

hemisphere showed a conspecific bias in the 14d Het-Env group. When 14d Het-Env subjects were directly compared to 4d and 9d Het-Env birds in a repeated measures ANCOVA (as above), there was a significant interaction of environment with hemisphere ($F(2, 430) = 8.0545$, $p = 0.000368$, Fig. 4a) with both the 4d and 9d Het-Env birds having higher ARMs in the left hemisphere contrary to the 14D Het-Env birds that had higher ARMs in the right hemisphere.

MUA responses were also analyzed to determine the adaptation rate (AdR) across individual presentations of the same stimulus. When normalized to remove the effects of response amplitude, AdR measures the normalized decrements in response amplitude as a function of stimulus repetition and, in naïve birds, reflect the familiarity of stimuli[37]: high adaptation rates are seen when a stimulus is novel and low rates indicate prior experience with a given stimulus. The present data show that normalized AdRs follow the same pattern as MUA responses (Fig. 4a, bottom panels). With exposure to Het-Env, they initially reverse to be faster (more negative) in the left hemisphere, and then return to the typical faster rate on the right side with prolonged exposure ($F(2, 430) = 2.779$, $p = 0.063$). These effects are not seen for Con-Env birds where AdRs remain consistently faster in the right hemisphere.

Lateralization patterns in ARMs remained right-biased when tested after a month of exposure to passive Con-Env or Het-Env playbacks. When 30d Het-Env birds were compared in a repeated measures ANCOVA to the 30 days Con-Env birds, there was no interaction of housing environment by hemisphere, indicating that patterns of lateralization were the same in both groups. Both groups had higher ARMs in the right hemisphere as indicated by a significant effect of hemisphere ($F(1, 367) = 16.861$, $p < 0.001$, Fig. 4b). Once again, there was a strong response bias for zebra finch stimuli ($F(1, 367) = 27.711$, $p < 0.001$) in both groups, consistent with previous findings. However, as seen in the 14 days cohorts, there was an interaction of hemisphere by stimulus type ($F(1, 367) = 4.9983$, $p = 0.023$) where the right hemisphere showed a conspecific bias and the left did not. When 4d and 9d Het-Env cohorts are included in the ANCOVA and compared against the 30d Het-Env group, there was a significant environment by hemisphere interaction ($F(2, 492) = 11.783$, $p < 0.001$, Fig. 4b) due to left hemisphere higher ARMs in the

4d and 9d Het-Env groups vs. right hemisphere higher ARMs in the 30d Het-Env group.

While the results suggested an overall conspecific bias (i.e. higher activity for conspecific vocalizations relative to heterospecific vocal signals) in both sets of analyses (Fig. 4a, b), Fig. 5 shows that similar changes in lateralized ARMs occurred for both canary and zebra finch test stimuli. Furthermore, conspecific bias was reduced in Het-Env ZFs in a hemisphere-dependent manner. At the 14D point, Het-Env cohorts displayed no significant differences in activity elicited by Zebra finch vs canary test stimuli within the left ($t(282) = -0.405$, $p = 0.343$) or right hemisphere ($t(282) = -1.178$, $p = 0.119$). Similarly, 30D Het-Env cohorts did not show significant differences in activity elicited by zebra finch vs canary stimuli within the left ($t(746) = -1.521$, $p = 0.064$); however, higher activity by Zebra finch, relative to canary, stimuli was observed in the right hemisphere ($t(746) = -2.045$, $p = 0.02$). Thus, conspecific bias was attenuated in the Het-Env for both hemispheres at 14D of exposure, and this pattern persisted in the 30D cohort in the left hemisphere.

In summary, birds exposed to Con-Env displayed typical patterns of right-biased NCM lateralization throughout the experiment and birds exposed to Het-Env displayed a transient shift in lateralized NCM activity and adaptation rate as a function of exposure to the novel acoustic environment. Interestingly, lateral differences in NCM activity were absent in the cohort exposed to silence for 5 days. Although the methods used in the acute experiment and the chronic ERP recordings were different, the temporal pattern of results of the former (Fig. 6) can be compared to a similar, but not identical, set of results in Fig. 2b. In both cases, the results are summarized in terms of *comparable* Lateralization Indices (LIs) across similar timescales and correspond in two ways. First, Fig. 6 contains NCM ARM analysis demonstrating that silence (Sil and Av-Sil cohorts) can attenuate lateralized responses, as was observed during the baseline recordings in Fig. 2b. Second, Fig. 6 shows a similar pattern of dynamic shifts in lateralized epidural activity, induced by Het-Env exposure, to that shown in Fig. 2b.

**Extended exposure facilitates auditory discrimination.** Zebra finches have been shown to be very poor at learning to discriminate between canary vocalizations in an operant task[44].

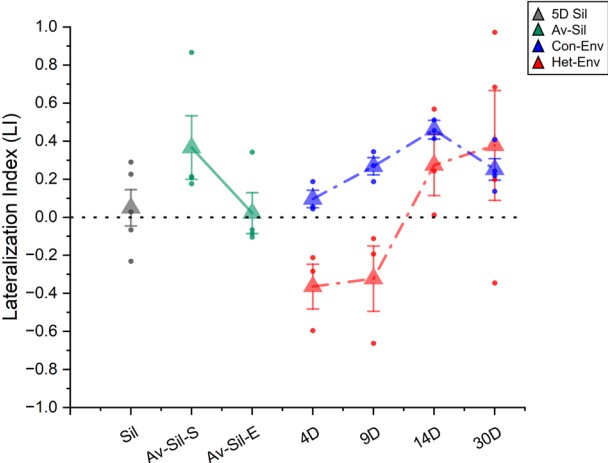

**Fig. 6 Summary of time-dependent changes in lateralized NCM activity as a function of acoustic experience.** Summary of MUA-derived ARMs as Lateralization Indices (LIs), as a function of exposure type (Silence, Het-Env, or Con-Env) and exposure duration (5D Silence, 4D, 9D, 14D, or 30D). Brief exposure to silence (5D Sil; grey) resulted in attenuated LIs, near 0 (i.e. bilateral activation). Birds recorded straight from the aviary (Av-Sil-S; green) displayed typical right-biased lateralization and a subsequent stay in silence for 5 days (Av-Sil-E; green) resulted in attenuated lateralization. On average, Het-Env exposure (red) for 4D or 9D resulted in left-biased (0<) NCM lateralization, while 14D or 30D exposure yielded right-biased (>0) NCM lateralization. Such transient shifts in lateralized activity were not observed in birds exposed to Con-Env (blue) for any exposure duration (4D, 9D, 14D, or 30D). Solid lines denote the same group tested at two timepoints and colored hashed lines denote independent groups of birds exposed to either Con-Env or Het-Env and separately sampled at each of the 4 exposure periods (4D, 9D, 14D, or 30D). Error bars denote 1.5 * standard error.

In order to assess possible behavioral benefits from exposure to canary vocalizations in Het-Env, we tested subjects in a GO-NoGO discrimination task after various periods of Con-Env and Het-Env exposure. As expected, Con-Env subjects performed poorly after all exposure durations (4, 9, 14 and 30 days); most failed to reach criterion even after extensive training (20+ days of GO-NoGO training). Furthermore, subjects trained after 9d exposure to Het-Env did not show significantly better performance than Con-Env subjects ($F(1, 8) = 1.6809$, $p = 0.23$, Fig. 7a). In contrast, subjects in the longer Het-Env exposure groups (14 and 30 days) had significantly higher final performance scores than Con-Env subjects ($F(1, 8) = 7.0740$, $p = 0.03$, Fig. 7b). With prolonged Het-Env exposure followed by operant training, 5 out of 5 subjects reached criterion of 80% correct, while only 1 out of 5 reached criterion in each of the three other groups (9d Het-Env, 9d Con-Env, and 14/30d Con-Env). Furthermore, of the subjects that reached 80% criterion, those with prolonged exposure to Het-Env reached criterion at 11 days on average while it took the single subjects in the other conditions 19, 16 and 21 days to hit criterion respectively.

## Discussion

Our neural data obtained with two different methods, chronic longitudinal ERPs and acute depth microelectrode recordings within NCM, show the same pattern of dynamic changes in lateralization. A few days' exposure to Het-Env reverses the typical pattern of hemispheric asymmetry: responses become larger on the left side relative to the right. However, after prolonged exposure to Het-Env, lateralization reverts to the typical right-biased pattern. These changes can be documented statistically by comparison with lateralization in Con-Env birds

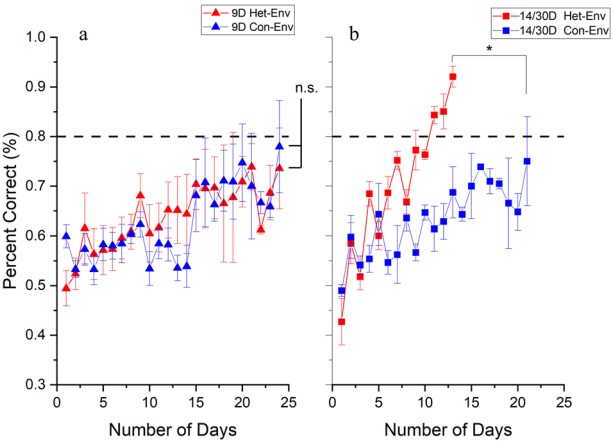

**Fig. 7 Performance in the behavioral discrimination of novel canary songs.** Go-NoGo performance in the discrimination of two novel Canary songs as a function of training days. **a** 9 days (9D) of Het-Env and Con-Env exposure (left panel) did not have an effect on discrimination performance after 24 days of operant training. **b** 14 and 30 days (14d and 30d; 14/30D) of Het-Env exposure (right panel) significantly facilitated discrimination performance but not in the cohort exposed to Con-Env. Error bars denote +/−1.5 s.e.m., (*) denotes significance at the 0.05 alpha level, "n.s." denotes non-significant differences, and the dashed line at 0.8 reflects 80% response criterion.

(Figs. 2b, 4). In both cases, birds are individually isolated and passively exposed to a novel environment; the Con-Env is an unfamiliar zebra finch aviary, while the Het-Env is an unfamiliar canary aviary. While canaries are also a species of finch, the acoustic patterns of their vocalizations differ from those of zebra finches (Fig. 1). Surprisingly, this acoustic difference, which can be characterized as a change in stimulus statistics, drives a dynamic change in the lateralization of auditory responses. More striking, this change is temporary and lateralization returns to the initial pattern after longer exposure.

Microelectrode recordings of MUA from NCM in both hemispheres demonstrate that significant changes occur there, while the ERP data reflect activity over much of the auditory lobule in the caudal forebrain, suggesting a wider distribution of these changes. Comparison of changes in more rostral with those in more caudal ERP electrodes found that the most caudal electrodes (i.e. in closer proximity to auditory areas) showed greater changes in lateralization as a function of exposure to novel sounds (Fig. 8); this supports the notion that the observed changes in lateralized activity are in part driven by auditory processing centers that include NCM, the caudal-most area in the auditory lobule.

Interestingly, adaptation rates (i.e. AdR) also undergo dynamic shifts which mirror the ARM-derived results. In Con-Env birds, as in naïve birds, novel canary and zebra finch stimuli elicit faster AdRs in the right NCM. This typical pattern is known to depend on normal rearing conditions in males[24]. We have previously shown that, in naïve birds pre-exposed to specific novel stimuli, both hemispheres show slower adaptation for those familiar stimuli than for novel sounds; this form of stimulus-specific adaptation reflects a long-lasting neuronal recognition memory for those sounds[37,39]. The faster left NCM AdRs seen in the 4d and 9d Het-Env cohorts appear to reflect increased participation of left NCM when the birds are faced with a dramatic change in the acoustic space; under such circumstances, the left NCM may become differentially engaged in the processing of novel stimuli. Note that the test stimuli differ from the environment stimuli in both cases. The typical lower rates in the left hemisphere

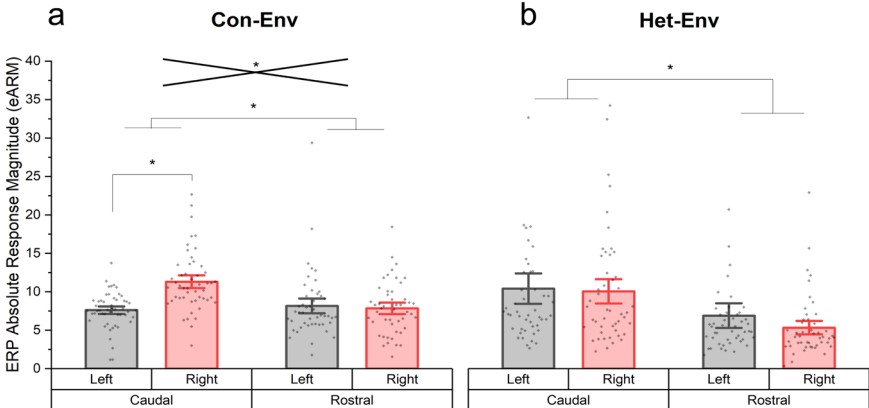

**Fig. 8 Comparison of epidural responses by hemisphere and position along the caudorostral axis.** Comparison of eARMs, between Het-Env and Con-Env, that were calculated from ERPs obtained from each hemisphere and classified caudal or rostral epidural-electrode pins. **a** The Con-Env cohort displayed higher eARMs in the caudal-most pins, relative to their rostral counterparts, suggesting that the caudal pins, which were located above the songbird auditory lobule, captured higher activity changes that were elicited by auditory stimulation. In addition, the caudal-most pins displayed lateralized dynamics that were not observed in the rostral-most pins (interaction denoted by the crossed lines). **b** The Het-Env cohort displayed higher eARMs in the caudal epidural pins relative to the rostral counterparts. Contrastingly, there was no interaction between hemisphere and electrode-pin position (i.e. rostrocaudal) and there were no lateralized differences in the caudal-most pins; likely due to the fact that this dataset was collapsed across time and therefore include timepoints in which birds display left- and right-lateralized activity which, when collapsed, appears as non-lateralized. Error bars denote $+/-1.5$ s.e.m., (*) denotes significance at the 0.05 alpha level, and intersecting lines denote a significant interaction between hemisphere (i.e. left versus right) and array-pin position (i.e. caudal versus rostral).

(of aviary and Con-Env birds) thus may be interpreted as familiarity with current, stable acoustic statistics. However, our experiment cannot disambiguate whether, within a given bird, it is the right NCM AdR that decreases or the left NCM AdRs which increases. Nevertheless, there appears to be a relationship between the MUA response magnitude and AdRs elicited by novel stimuli within a given hemisphere, which covary when ZFs experience a dramatic change in the acoustic environment. Note that AdR are normalized to activity levels at each site, so the rate changes are truly independent of MUA response magnitude.

The initial reversal of lateralization for both MUA responses (i.e. ARM) and AdR in 4d and 9d Het-Env groups is likely driven by the novelty of the acoustic statistics in canary vocalizations. This leads to the idea that, after 14d of Het-Env exposure, the stimulus statistics of canary stimuli have become familiar and no longer pose a novelty challenge to the auditory system, so it returns to its baseline state (right-biased). However, the brain has changed the way it processes these stimuli through a mechanism that involves different changes in NCM in the two hemispheres. The typical preference for conspecific song is lost in left but not right NCM, a persistent effect. Of greatest importance are the behavioral results: enhanced discrimination performance for canary sounds is found only in birds that have experienced prolonged (14d and/or 30d) Het-Env (canary) exposure (Fig. 7), at a time when lateralization is presumed to have reversed and returned to baseline. We do not have direct electrophysiological confirmation of lateralization shifts in these Het-Env birds, but the following discussion assumes that the dynamic asymmetries documented after various periods of Het-Env exposure occurred in those birds as a group at the same time points.

The dynamic component of lateralization is proposed to reflect, and perhaps facilitate, the incorporation of novel stimulus-feature statistics into a neural circuit containing acoustic representations that are accessed to serve behavioral interactions with the environment. Naïve Zebra finches have a response bias for conspecific vocalizations and are much better at discriminating two conspecific vocalizations than two heterospecific vocalizations[44]. However, our data show that this behavioral limit can be overcome by passively exposing subjects to the novel acoustic category

prior to operant training that tests discrimination. We thus interpret our combined results as evidence that the lateralization shifts elicited by prolonged exposure to Het-Env subserve improved behavioral discrimination of heterospecific auditory stimuli. The return to lower MUA responses and slower adaptation in left NCM could reflect that side's participation in recognition memory for the class of canary sounds, by analogy to stimulus-specific adaptation that reduces ARMs for familiar individual songs[37]. The reduced response bias for conspecific relative to heterospecific song in the left hemisphere but not the right is also consistent with this picture. In summary, the return to typical patterns of lateralization (for both MUA responses and AdRs) would thus reflect memory consolidation for the novel stimulus statistics, and/or a stabilization of newly formed connections which are now available for use in discrimination behavior. This interpretation implies that sequential exposure to multiple novel acoustic environments (e.g. Het-Env-1, Het-Env-2, Het-Env-3, etc…, each for ~14 days) should elicit transient shifts in asymmetric activity that reflect the reorganization of the neural representation of the acoustic statistics for each environment. Results from this type of experiment will strengthen the view that novel acoustic statistics drive the process of dynamic changes we measure as the reversal and return of lateralized activity.

Surprising additional evidence for the lability of lateralization was found in birds exposed to a period of silence during the ERP baseline recordings: they did not show typical, right-biased patterns of activity. This observation was further assessed in two separate cohorts via microelectrode electrophysiology in NCM. The data show that NCM activity is no longer lateralized in birds that spend 4–5 days in silent isolation (hearing only themselves). While it is difficult to interpret the functional role of the loss of lateralization in auditory areas that are deprived of stimulation, it is possible that other biological processes are being engaged by such an experience: e.g. in ZFs, a single day of silence and social isolation can drive changes in gene activity, prompted by the engagement of epigenetic mechanisms[45]. In any case, we can conclude that silent isolation should be considered an auditory manipulation; it cannot be treated as a baseline or control condition in future experiments.

Taken together, the pattern of results observed as a function of exposure to silence, heterospecific, conspecific, and aviary environments suggests that lateralized activity patterns are not static and can be modulated by recent experiences even in adulthood. A more striking interpretation follows from the results in cohorts sampled after exposure to silence. Lateralization is not only modifiable but its existence depends on continuous auditory inputs. We have documented that: (1) silence *abolishes* asymmetry in responses; (2) continued exposure to familiar acoustic environments *maintains* typical asymmetric response patterns; and (3) exposure to novel acoustic environments drives a *transient reversal* of typical lateralization that then returns to the typical pattern after a prolonged period of exposure. In sum, what we consider the "typical" right-biased responses are not fixed, but actually reflect the state of a bird who is hearing a familiar acoustic environment. In this view, the typical pattern of lateralization is simply a snapshot of the normal state for ZFs living in a conspecific aviary, familiar with ZF acoustics and unchallenged by foreign sounds.

Taken together with the observed improvement in behavioral discrimination, these findings suggest that the dynamic changes in lateralized activity are not epiphenomenal but are evidence of a process of functional reorganization. We hypothesize that this involves the creation of new perceptual filters, and that these new filters, built or amended during passive exposure to canary sounds, then detect acoustic features of those sounds that enable improved behavioral discrimination. Of course, we do not know precisely where or how this occurs, but lateralized activity in NCM at least provides a read-out of this process.

Our observations that typically right-lateralized[16,24] auditory responses are modifiable in adult ZFs are even more striking considering that *aves*' brains do not contain a corpus callosum (CC); the hemispheres are instead connected via the anterior commissure (AC). Functional lateralization for language is correlated with midsagittal surface area of the human CC which may contribute directly, or indirectly, to previously observed changes in lateralized activity observed in humans learning a new language[46]. A recent imaging study provided evidence of coordinated activity between the songbird hemispheres and suggested the AC as a possible pathway[47]. Nonetheless, the role of the AC in hemispheric communication that serves auditory processing and its possible involvement in the plasticity we describe here remains unknown. The contribution of contralateral projections at the thalamic, mesencephalic and brainstem levels cannot be ruled out. For example, it is possible that transient shifts in lateralized activity of ZFs exposed to Het-Env are explained by midbrain or thalamic gating of auditory upstream processing and/or due to the current state of processing by each hemisphere during Het-Env exposure. Future studies should aim to discern the role of cortico-cortical projections (i.e. CC; pallio-pallial via AC in the songbird) on lateralized activity in the context of second language learning (cf. passive exposure to novel stimulus statistics in ZFs) or the degree to which mesencephalic and/or thalamic structures may gate interhemispheric activity through possible reciprocal connections in the afferent processing stream.

Our results are based on ERP and microelectrode measurements that show relative differences between the two hemispheres in response magnitude and adaptation rate. While suggestive of circuit level changes, the present methods do not directly assess events at the cellular or circuit level. Nonetheless, a consideration of known NCM response properties suggests some possibilities. For example, as shown, passive exposure to the canary Het-Env for 14d or 30d is sufficient to improve operant discrimination of two novel canary songs, but 9d of exposure is not, suggesting a fast change in perceptual discrimination between 9d and 14d days that underlies improved operant discrimination. Neural processes

that occur on the time scale of 14d could include migration and incorporation of new neurons into functional circuits (20-30 days). We know that ongoing neurogenesis in NCM favors the left side[41] and could contribute to heightened levels of plasticity[48,49]. In rats, exposure to an enriched environment has been shown to increase survival of immature neurons[50]. We speculate that novel sounds might recruit more new neurons in the left hemisphere which, combined with the higher excitability of immature neurons[50], would contribute to the higher ARMs seen on the left side in 4d and 9d Het-Env groups. Furthermore, Zebra finches that were faster learners in a conspecific GO-NoGO discrimination task displayed a higher degree of left-biased activity and faster adaptation rates the end of 14–21 days of training[51]. If more new neurons are rescued as a result of activity driven by novel auditory environment exposure, they could integrate into existing circuits and contribute to plasticity that underlies overall discrimination of both conspecific zebra finch and heterospecific canary stimuli. In this view, as these rescued neurons mature over the course of 14–30 days, their excitability decreases, bringing ARMs in the left hemisphere down to the normal levels seen in our 14d and 30d Het-Env groups.

Lateralized sensory processing for acoustic communication signals is seen in many vertebrates, including songbirds, bats, and non-human primates[25,26,52]. Thus, the origins of lateralized processing for human speech may exploit these evolutionary precursors. The asymmetry in the vertebrate nervous system has been described in terms of differences in the time scale of processing longer time windows (lower temporal resolution) in the right and shorter time windows (higher resolution) in the left hemisphere (Asymmetric sampling in time, AST[53]); e.g. spectrotemporal features of vocal signals that occur in longer timescales (i.e. prosody) are present in vocalizations of young animals[54,55] and typically thought to be processed in the right hemisphere in mammals. In addition, ZFs seem to have a propensity for processing and categorizing sounds based on their spectrotemporal content[56–58] which may be consistent with their typical right-lateralized processing of conspecific song (cf. higher right-lateralized activity in humans processing prosodic cues). While it is possible that the acoustic characteristics of canary song somehow engage with these differences to drive the reversal since canary songs have higher frequency content, this argument is not consistent with the observation that responses to ZF test stimuli also reverse.

More generally, it remains unknown how the processing of fast- and slow-frequency stimulus content is distributed between the two hemispheres in zebra finches, so it is unclear how to align the AST frameworks with ZF lateralization, and there is no simple explanation for the dynamic reversal in terms of the frequency content of heterospecific sounds. Novelty must be invoked as an essential part of the explanation. The reported shifts in lateralized activity may thus reflect a shift in the resolution at which stimuli are processed when the ZF is challenged by a dramatic change in the acoustic environment. We may speculate that the accommodation of novel stimulus statistics requires a transient shift to higher sampling resolutions (left-lateralized activity), followed by a return to lower sampling resolution (*native*/typical right-lateralized activity) once perceptual filters have been updated to encode the novel stimulus statistics. While there is no direct evidence for this explanation, it is presented as an extension of the AST framework, whereby the time-sampling-dependent lateralized processing of communication signals may be contingent on familiarity and/or passive learning during exposure to novel sounds. Such an interpretation may also apply to humans learning the sounds of a new language.

Studies in humans have shown that patterns of lateralization can reflect the history of language acquisition and proficiency.

In bilinguals, the second language is lateralized in the same hemisphere as the native language if acquired early in life, but bilaterally activated if acquired later in life[59]. Although lateralization patterns between the two groups differ due to age of acquisition, pattern differences might also reflect language proficiency, where later learners are less proficient than earlier learners. In a human study where an artificial grammar was taught to adults, representation of the new grammar changed from bilateral to left lateralized over the course of training as proficiency increased[46]. Furthermore, since the human subjects were sampled for lateralization in adulthood, possible *changes* in lateralization over the course of earlier natural learning are unknown. Changes in patterns of lateralization could be an indicator of novel category acquisition, both in development and adulthood, and may predict improved behavioral discrimination of novel category sounds. Our data do not mirror these results exactly, as our birds showed an initial reversal of lateralization, rather than bilateral activation after 4d and 9d exposure to Het-Env. However, the return to a normal pattern of lateralization after 14d and 30d in Con-Env in our birds is also correlated with proficiency as demonstrated by better performance in the operant discrimination paradigm. Thus, our data suggest a general relationship between lateralization dynamics and perceptual competence with a novel stimulus class[2].

As seen in our current and earlier[24] results, ZFs that have been living in a noisy conspecific aviary show higher activity in the right NCM when passively hearing conspecific vocalizations, and we also show that this pattern depends on sustained stimulation. Although lateralized brain activity that gives a right field (left hemisphere) advantage for sensory processing seems universal across vertebrates, including some birds[60], the ZF pattern of auditory responses appears to be the reverse (manifestations observed as early as the songbird thalamus, MLd[61]) of the typical characterization of speech-related functions as left-lateralized in human listeners. While it is tempting to see speech and song as somewhat equivalent communication signals, and NCM as roughly analogous to Wernicke's area (due to preferential responses to conspecific sounds), there are relevant differences in methodology and context between human speech studies and our ZF paradigm that may contribute to this apparent difference. While left-lateralized activity is seen for connected speech (and is most pronounced for processing sentences), isolated speech sounds show bilateral activity in human imaging studies[62]; top-down processing that reflects syntactic and semantic processes may drive left lateralized activity in temporal cortex. Active human speech tasks (and even involuntary semantic processes during passive listening) may engage the brain in ways that passive playbacks fail to do in ZFs or in humans[63]. In the current ZF experiments, left-biased activity elicited by exposure to foreign sounds may be due to attentional mechanisms not engaged by familiar ZF vocalizations. Finally, spectro-temporal features of the stimuli may contribute to asymmetric activity, since the left hemisphere in humans seems tuned to rapid temporal transitions that distinguish phonemes, while the right side responds to slower modulations and spectral content. Although ZF songs have a prominent rhythm of syllables and silences, behavioral testing reveals that spectral features may be more salient[56,57], features that may preferentially elicit right hemisphere activity if ZFs show a similar asymmetry to humans.

Regardless of whether the aforementioned discrepancy is due to methodological differences (e.g. passive playback vs. task demands[64]), the spectrotemporal content of communication signals, or an essential species difference, our work provides clear evidence for a functional division of labor when the ZF brain is challenged by a foreign distribution of acoustic features. These dramatic changes in the stimulus statistics of the environment (e.g. exposure to a novel acoustic environment or a second language that contain acoustic stimuli with novel temporal and spectral features) may engage dynamic and lateralized neuroplasticity mechanisms that allow juvenile and adult organisms to adapt when challenged with novelty. Furthering the understanding of the time course and triggers of auditory plasticity in the mature adult brain, can have important implications for therapies that address hearing recovery and improving strategies that are conducive to second language acquisition.

## Methods

**Animals and housing.** Adult male zebra finches (aged >120 phd) were randomly sampled from our aviary at Rutgers University or purchased from a local supplier for all of the experiments carried out in the current study. All birds were healthy, had undergone normal rearing conditions, and were maintained on a 12/12-h light cycle (7am–7pm) with *ad libitum* food and water. During passive exposure to acoustic environments or silence, birds were individually housed in sound-proof isolation boxes (83.82 cm × 40.64 cm × 45.72 cm) that contained a speaker. Of the total birds used in the series of experiments, a subset was randomly assigned to be passively exposed to either a novel heterospecific ($n = 5$) or a novel conspecific ($n = 5$) acoustic environment for different periods of time to assess the time course of dynamic lateralization via chronic electrophysiological recordings of Event-Related Potentials (ERPs). To determine the effect of silence on lateralized patterns of NCM activation, initially observed in the baseline epidural data of the aforementioned cohort, two approaches were used: (1) One cohort ($n = 7$; Sil) was removed from the aviary, placed in silence for 5 days, and recorded, and (2) another cohort ($n = 4$; Av-Sil) underwent two separate acute recordings, initially straight out of the main aviary (Av-Sil-S) and then following 5 days of silence (Av-Sil-E). A separate cohort of Zebra finches was exposed to either conspecific or heterospecific environments for different periods of time, followed by bilateral microelectrode recordings of Multi-Unit Activity (MUA) from NCM (MUA obtained from a total sample of $n = 25$). Finally, to determine the extent of exposure needed to facilitate auditory discrimination of novel heterospecific exemplars, a behavioral performance assay was conducted in an additional set of birds ($n = 5$ in each experimental condition, $n = 30$). Surgeries, electrophysiology recordings, and behavioral assays employed in this study complied with ethical regulations for animal testing.

**Acoustic environment manipulation.** The passive exposure paradigm employed in the present set of experiments was adopted from a previous study in which Zebra finches experienced either a novel conspecific acoustic environment (playback of a 12 h recording of an unfamiliar Zebra Finch aviary; Con-Env) or a novel heterospecific acoustic environment (playback of a 12 h recording of an unfamiliar canary aviary; Het-Env) for various time periods[16]. Playback environments were recorded over 12 h periods from zebra finch and canary aviaries at the Rockefeller University Field Research Station (in Millbrook, NY) to ensure that all acoustic environments were novel at the start of the experiment. Exposure of the Con-Env group to zebra finch songs and calls represented the 'native' acoustic environment. Exposure to canary songs and calls in the Het-Env group provided a 'foreign' acoustic environment. Het-Env stimuli contain relatively more energy at higher frequencies and longer durations than Con-Env stimuli (Fig. 1a). The Het-Env sounds presented contained acoustic features typical of canary vocalizations such as rapid trills and high pitched whistles with long durations and many syllable repetitions. In contrast, Con-Env sounds contained acoustic features commonly found in zebra finch vocalizations

such as broadband harmonic stacks, fast frequency modulations, and shorter syllables (Fig. 1b). Vocalizations in the two environments occupied distinct, but partially overlapping, sectors of acoustic space, such that Con-Env and Het-Env present two separate constellations of sounds, both of which lie within the perceptual range of the zebra finch[65].

In the current experiment, these two environment recordings were continuously played throughout the entire lights-on period (7am–7 pm). Sound intensities for the two environments were matched to each other and to the sound intensity in the general zebra finch aviary and were well below the threshold for causing peripheral damage[66]. Playback amplitude at the end of the cage closest to the speaker averaged 67 dB SPL (A-scale) with occasional brief peaks not exceeding 74 dB SPL (A-scale, fast) in both Con-Env and Het-Env. In the case of the Sil and Av-Sil groups (Av-Sil-S and Av-Sil-E), exposure to silence consisted of placing birds in social and acoustic isolation for 5 days; the birds were allowed to self-vocalize. For all auditory- and silence- exposures, ZFs were single housed in sound-attenuating chambers.

To determine the time-course of lateralization changes as a function of acoustic exposure and time, two different electrophysiological methods were used (see detail below). In one set of birds, chronic epidural recordings were obtained every other day over 20–22 days of passive sound exposure. In a second set of birds, bilateral NCM activity was assessed with acute microelectrode recordings that followed different exposure periods and acoustic environments. Subjects were randomly assigned to one of 8 groups (Het-Env for 4d, 9d, 14d or 30d or Con-Env for 4d, 9d, 14d or 30d). An additional set of birds was passively exposed to either Con-Env or Het-Env for 9, 14, or 30 consecutive days followed by a Go/No-Go operant conditioning paradigm. All procedures conformed to a protocol approved by the Animal Care and Use Committee at Rutgers University.

**Epidural electrode array**. The epidural electrode array consisted of a modified 10-position, dual-row, Nano-strip connector (NPD-10-DD-GS, Omnetics). This connector mated with the corresponding connector of a miniature 9-channel headstage pre-amplifier (Neuralynx HS-8-CNR-MDR50) that was used to record neural activity. A short (1.5 mm) silver ground wire was soldered to the male pin of the Nano-strip that aligned with the reference pin of the headstage. The remaining 8 male pins (2 rows of 4) of the connector array were shortened to a length of 2 mm and were bent outwards (from the base) such that the tips of each row of 4 pins was located 1.1 mm from the center. This allowed each row of 4 pins to make contact with the dura, atop each hemisphere independently, when implanted.

**Pinning surgery**. Zebra finches were anesthetized using 1–3% Isoflurane in oxygen, placed in a stereotaxic instrument, and head-fixed with ear bars and a beak-bite. A head fixation pin was cemented onto the anterior portion of the skull to stabilize the bird's head during electrophysiology. Next, a craniotomy was performed to expose the caudal telencephalon and the midsagittal sinus. A chamber was formed from dental cement (Anhydrous Polycarboxylate Cement; Tylok Plus, LOT: 161018) around the craniotomy. As described in the following sections, different procedures were employed following the pin implantation to allow for the different electrophysiology assessments (i.e. ERP or MUA). Post-operation, Zebra finches were supervised until full recovery (30–60 min) from the short-acting anesthetic[67]. Next, the birds used to chronically record ERPs were returned to the general aviary for ~48 h (after array implantation) and the birds used to assess MUA were returned to their respective experimental conditions.

**Electrode-array implantation and ERP electrophysiology**. Post-pin implantation, and while the bird remained anesthetized, the epidural array was lowered onto the dura of the brain with its long axis centered over the midsagittal sinus; the caudal-most pins on each side spanned the bifurcation of the sinus. The silver ground wire was tucked between the dura and the inner layer of skull on the left hemisphere (Fig. 9a, b). Silastic compound (Silicone Elastomer, Kwik-Cast) was used to secure the position of the pins, to isolate them from each other, and to fill the void in the chamber. Dental cement was used to secure the microarray onto the chamber on the birds' head. Anesthesia was discontinued and the subjects were then removed from the stereotaxic apparatus and allowed to recover under a heat light until they perched, ate, and drank (30–60 min).

For bilateral epidural recordings, each bird was temporarily moved from the environment exposure cage a large soundproof booth (IAC Inc., Bronx, NY). The bird was comfortably restrained in a custom plastic tube, the head fixation pin was clamped to a stereotaxic frame, and the headstage pre-amplifier was connected to the implanted epidural electrode array. Electrical signals were amplified with a gain of x1000, filtered (bandpass 1-1000 Hz), and digitized with a Power 1401 A/D converter (Cambridge Electronic Design) by using Spike2 v7.01 software and custom scripts to record and further process the waveforms. Recordings were obtained in test sessions lasting ~30 min conducted every day during the baseline period (4 days), when the subjects were isolated in silence, and every other day during the exposure phase (20–22 days) when the subjects passively heard either the novel heterospecific (i.e. Het-Env) or novel conspecific acoustic (i.e. Con-Env) environments.

**Regional source of auditory ERPs**. Event-related potentials (ERPs; Fig. 9c) were used to calculate an index of activity strength (eARMs; see Methods) captured by each electrode-pin of the epidural array. To confirm that ERPs originated from auditory areas, eARMs magnitudes and adaptation profiles were analyzed between the caudal- and rostral-most electrode pins (caudal and hemisphere pin assignment was defined by quadrants divided by the sagittal and caudo-rostral axes; each quadrant included 2 pins and represented one hemisphere and caudal/rostral position); the caudal-most pins were located on the portion of the dura that covered the songbird auditory lobule (see Methods; Fig. 9a, b).

To determine the regional and lateralized influences on auditory ERP magnitudes, eARMs were plotted as a function of hemisphere and epidural pin position (caudal vs rostral), irrespective of time. Figure 8 suggests that, when the analysis was conducted for each exposure group separately (Con-Env: Fig. 8a, Het-Env: Fig. 8b), the caudal-most pins capture higher eARMs (Con-Env: $F(1, 192) = 8.1$, $p = 0.004$; Het-Env: $F(1, 188) = 15.64$, $p < 0.001$). There was an interaction between hemisphere and pin position ($F(1, 192) = 15.487$, $p < 0.001$), which was mainly driven by the right-lateralized eARMs ($t(192) = 5.201$, $p < 0.001$) in the caudal pins. Contrastingly, the Het-Env group showed no such interaction ($F(1, 188) = 0.343$, $p = 0.558$) and did not show any lateralized eARMs in the caudal epidural pins ($t(188) = -0.236$, $p = 0.41$), likely because the data was averaged over periods where lateralized activity was shifting between hemispheres and therefore would appear as a non-significant difference between left and right eARMs in the time-collapsed graph.

Similar to the known stimulus-specific adaptation phenomenon observed in NCM[37], whereby electrophysiological responses adapt as a function of stimulus repetition, there were instances in which eARM responses were observed to decrease as a function of stimulus repetition within a recording session. While this

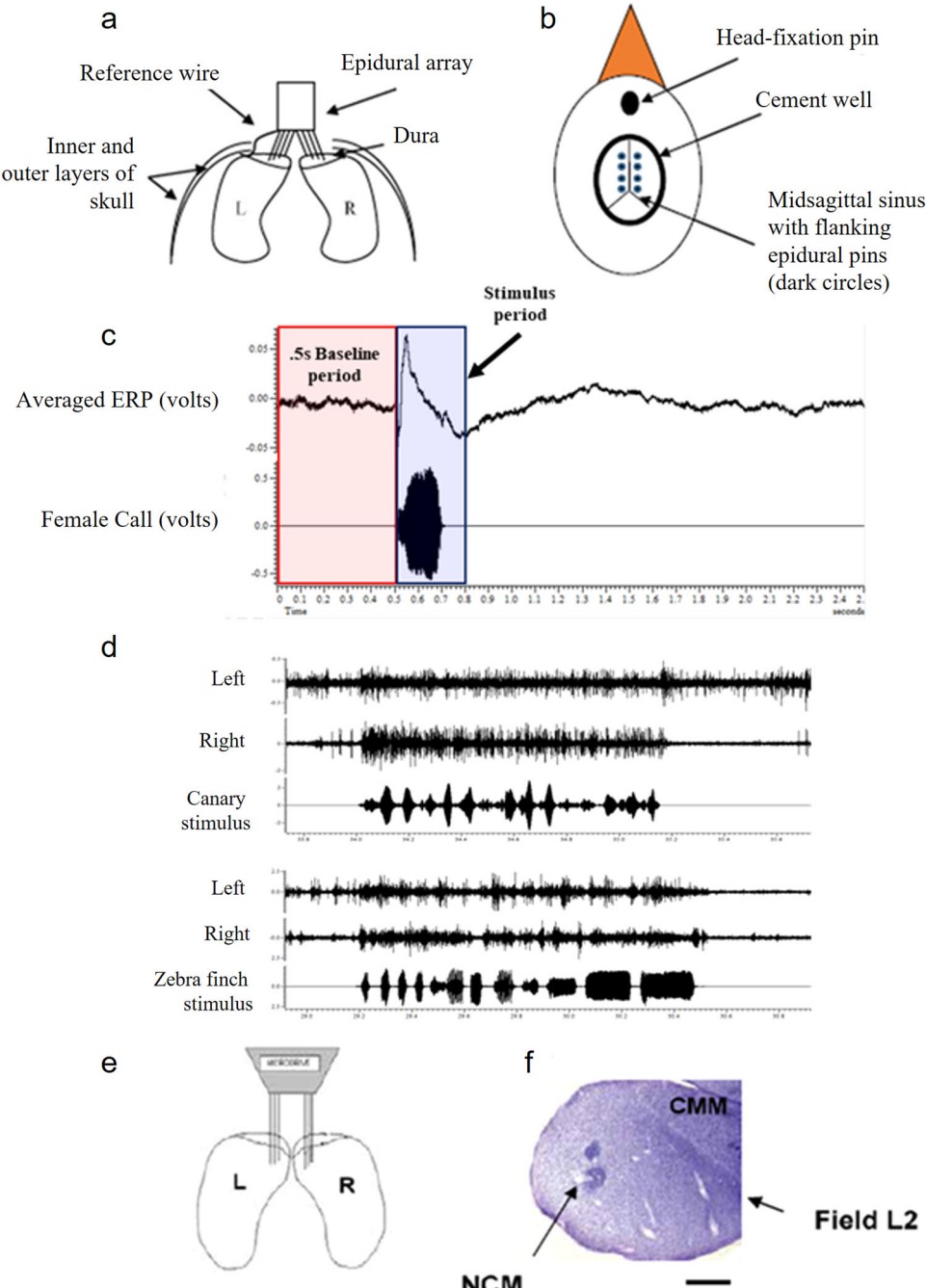

**Fig. 9 Illustration of epidural array and acute microelectrode positioning, and examples of epidural and multiunit activity.** Epidural electrode array implant (shown without the cement well for illustration purposes), NCM anatomy, and chronic and acute electrophysiology. Epidural electrode array implant (shown without the cement well for illustration purposes) and electrophysiology. **a** Eight electrode-pins (4 on each hemisphere: 4 caudal and 4 rostral) were lowered onto the dura and parallel to the midsagittal sinus. **b** The bifurcation of the midsagittal sinus was used as a reference to determine the position of the array by flanking it with the caudal-most electrodes. A ground/reference silver wire that was soldered onto the caudal-most pin from the left array column was tucked in between the dura and the inner layer of skull. **c** Sample ERP and stimulus (volts) as a function of time (seconds). The red window reflects a 0.5 s baseline period and the blue window reflects the stimulus period plus a 100 ms offset, both of which were employed during ARM caluclations. **d** Raw electrophysiological recording from electrodes placed in left and right NCM showing bilateral responses to canary (top) and zebra finch (bottom) test stimuli. X-axis time bar: 0.5 s. Y-axis scale bar: 25 µV. **e** Schematic in coronal view of bilateral electrode placement in NCM. **f** Sagittal section of NCM stained with cresyl violet with histological confirmation of electrode placement, black arrows indicate location of 2 electrolytic lesions made in NCM at the conclusion of recording; scale bar: 1 mm. Labels show locations of NCM, CMM and the primary auditory area, Field L2. Figure 9e, f were adapted from Fig. 3a, b in Yang and Vicario, 2015, Neuroscience, Copyright Elsevier[16].

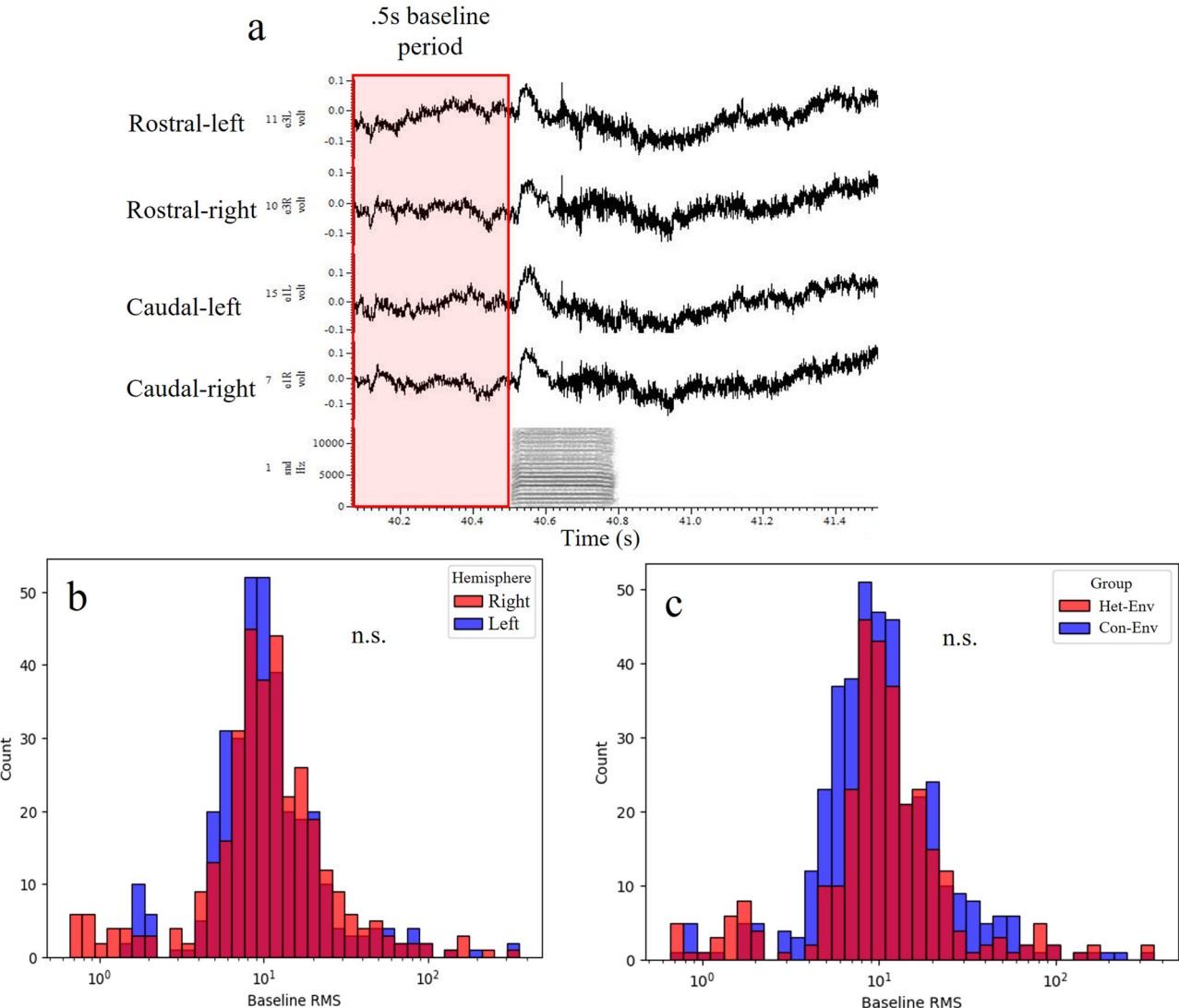

**Fig. 10 Analysis of baseline epidural activity between hemispheres and as a function acoustic experience.** Sample of raw epidural waveforms and statistical summary of baseline levels of activity. **a** example of raw epidural voltage signals for four different epidural array electrode-pins (2 from the right and 2 from the left hemisphere; 2 rostral and 2 caudal). Note that the baseline voltage level is similar across channels and there is a characteristic ERP in response to a female call stimulus (bottom channel, sonogram). For all analyses, epidural waveforms are averaged by stimulus and eARM is calculated as the difference between the root mean square (rms) during stimulus and the baseline rms; the 0.5 s baseline period sampling window is indicated by an overlayed red rectangle, baseline activity was used for the analyses shown in panels (**b**, **c**). **b** Distribution of baseline RMS values between left and right epidural pins are not statistically different suggesting that there was no systematic effect of the left-hemisphere positioned reference wire. **c** Distribution of baseline RMS values between Het-Env and Con-Env exposure groups were not statistically significant, suggesting that auditory exposure did not affect baseline epidural activity and therefore there was no systematic effect of the reference wire on exposure-specific effects. "n.s." denotes non-significant differences.

supported that ERPs were driven, in part, by NCM activity, detailed analyses of eARM adaptation were not carried out because the variability of ERPs requires averaging many trials, which makes precise analysis of adaptation across trials difficult.

As seen in Fig. 9a, the epidural array contained a reference wire that was placed atop the left hemisphere, for all finches, away from the recording epidural pins. Therefore, a *post-hoc* analysis was conducted to determine whether there were systematic influences of reference-wire placement on our measurement of lateralized epidural activity; e.g. lower distances between recording and reference electrodes can lead to the detection of lower voltage signals by the recording electrode[68]. Therefore, if indeed there was an influence of the left-lateralized ground wire on detected voltage levels, caudal-left pins would exhibit lower activity during baseline. Figure 10a illustrates 4 raw epidural

traces of electrical activity for epidural pins along the rostrocaudal axis and hemispheres; qualitatively, there were no differences in the baseline level of recorded activity (all near 0). To quantitatively determine the effect of ground wire placement on epidural activity, rms was calculated for the baseline period (brms; 0.5 s prior to stimulus onset) for every recording; brms is a measurement used in the eARM calculation. Figure 10b–c illustrates the distributions of baseline rms as a function of hemisphere or exposure group. The distributions did not differ significantly between hemispheres (Fig. 10b, $H = 0.718$, $p = 0.397$) or groups (Fig. 10c, $H = 1.301$, $p = 0.254$). Together with Fig. 8, we interpret the results as the lack of systematic influence of reference-wire positioning on our measurement of lateralized activity. In any case, the placement of the reference wire on the left cannot explain why lateralization shifted over

time in the same bird exposed to Het-Env. Therefore, the results shown in Fig. 2 are most likely explained by auditory exposure.

**Multi-unit activity (MUA) electrophysiology.** At the end of their respective exposure periods and 48 h post-pin-implantation surgery, subjects were removed from isolation boxes and placed in a soundproof booth (IAC Inc., Bronx, NY) for electrophysiological recording. Subjects were restrained in a custom made plastic tube and head-fixed to the stereotaxic apparatus using the previously implanted head pin. A multi-electrode microdrive (Thomas Recording, Giessen, Germany) was used to place 16 tungsten micro-electrodes (Type ESI2ec, impedance: 2–4 M ohm, Thomas Recording) in NCM bilaterally (8 in each hemisphere). Coordinates used for placement of micro electrodes within the boundaries of NCM were 0.5–1.5 mm rostral and 0–1 mm lateral to the bifurcation of the mid-sagittal sinus (the zero point for songbird stereotaxis). Each electrode was lowered into the brain while white noise stimuli with the amplitude envelope of canary song were played to identify the first responsive site along the track. Once responsive sites were located on all electrodes, playback of the testing stimuli commenced. Multi-unit recordings of neural spike activity (Fig. 9d, e) were taken simultaneously from all electrodes using Spike2 v7.01 software (CED, Cambridge, England). Recorded activity was amplified (x19,000) and band-pass filtered from 0.5-5 kHz.

**Auditory testing stimuli for ERP electrophysiology.** Natural female zebra finch long calls were used as acoustic stimuli to assess lateralized activity, due to the ethological relevance of the stimulus and the heightened responsiveness seen in Zebra finch males that hear a female call[69]. In addition, female calls were employed because: (1) song stimuli with multiple syllables (lasting 500–1000 mS) elicit highly fluctuating ERPs that are difficult to interpret and process (contrary to calls, songs contain inherent silences that produce ERP waveforms that are not discrete), (2) female calls limit the influence of auditory exposure to complex conspecific vocalizations (i.e. zebra finch songs as test stimuli) that might themselves elicit changes in lateralization, and (3) female calls are simpler zebra finch vocalizations that are sufficient to drive lateralized auditory responses[24].Stimulus sets consisted of 3 female calls (180–300 ms duration; sampling rate: 44 kHz) each presented 100 times in shuffled order (total 300 stimuli presentations per testing session), with an inter-stimulus interval of 5 s. Stimuli were played from a midline speaker 0.5 m in front of the subject at an intensity of 70 dB SPL. At each test session in time, each subject heard a set of 3 novel calls to avoid familiarity effects. The order of stimulus sets was randomized between subjects to account for order and stimulus-specific effects. An example of a call stimulus and the resulting ERP average is shown in Fig. 9c.

**Auditory testing stimuli for MUA electrophysiology.** Auditory stimulus sets consisted of 5 novel zebra finch (zfstim) and 5 novel canary songs (canstim) that were not part of either playback environment. A separate set of novel stimuli was used at each group of simultaneously recorded sites. In each set, the 25 repetitions of each of the 10 songs (duration: 800–1500 ms; sampling rate: 40 KHz) were presented in shuffled order at 8 s ISI. All song stimuli were played back at 70 dB through a speaker centered 30 cm in front of the subject. Each set of song stimuli was followed by presentation a set of pure tone stimuli to obtain tuning curves and best frequencies for each recording site. Tone stimuli ranged from 500 Hz to 5000 Hz at 250 Hz increments (20 stimuli; duration: 260 ms; sampling rate: 40 KHz; 3 repetitions each in shuffled order; 6 s ISI). After presentation of both song and tone

stimuli, all electrodes were lowered 300 μm along the dorsal-ventral axis and a new set of song stimuli was played, followed by the tone set. This was repeated until all 4 sets of novel songs had been played at their respective depths in NCM. Upon completion of testing, electrolytic lesions (10 μA for 10 s) were made at the final recording depth in each hemisphere for histological verification that sites were within the boundaries of NCM.

**Histology following MUA electrophysiology.** Three days post-test, subjects were deeply anesthetized with Nembutal and transcardially perfused with saline and 4% paraformaldehyde. Brains were removed and post-fixed in 4% paraformaldehyde. Sagittal sections (50 μm) were taken through left and right NCM on a Vibratome, stained with cresyl violet and visualized under light microscopy to identify lesioned recording sites (Fig. 9f). Sites determined to be outside of the histological boundaries of NCM were excluded.

**Quantification of neural activity.** ERPs and MUA were used to assess differences in neural activity between hemispheres and groups, by quantifying response magnitudes and the rate of adaptation following repeated presentation of the same stimulus. These measures have been shown to differ in a lateralized manner in NCM[24] and the pattern of lateralization in these measures is dependent on the extent of novel experiences[16].

MUA Absolute Response Magnitude (ARM) measures the strength of neural responses, via MUA, to auditory stimuli and is calculated as follows: the root mean square (rms) of neural activity during the control period (500 ms prior to stimulus onset) was subtracted from the rms of the stimulus-on period (stimulus duration plus 100 ms), following established methods in the laboratory[24,39]. ARMs were averaged for trials 2–6 of each song stimulus and each stimulus category (zfstim or canstim) to yield a response strength for conspecific songs and a response strength for heterospecific songs for each recording site to be used in further analysis.

Adaptation Rate (AdR) measures the normalized rate of decrease in response size to a repeated stimulus[37,39]. Adaptation rate is calculated as the slope of the regression line over repeated presentations for trials 6–25 with a given stimulus divided by the average ARM of those trials. Dividing by the average ARM normalizes for differences in absolute response strength between recording sites. Thus, adaptation rate is the percentage drop in response amplitude per stimulus repetition at each recording site. In NCM, repeated novel stimuli elicit robust responses which rapidly adapt, while responses to familiar stimuli are already adapted and only undergo minimal additional adaptation. Thus, high adaptation rates signify novelty, while low rates indicate familiarity for a stimulus - a kind of long-lasting neuronal memory[37]. Adaptation rates were used to sort the MUA data: only those sites in NCM that showed adaptation profiles (see Data Analysis, below) characteristic of NCM neurons were included in the analyses. The vast majority of neurons and sites in NCM are known to exhibit stimulus specific adaptation, while those in the adjacent primary auditory area, Field L, do not[37]. Thus, we eliminated recording sites that did not meet the adaptation criterion (AdR < −0.05) established in earlier work[39]. Across the different groups, 10–12% of sites were excluded based on this criterion.

ERP Absolute Response Magnitude (eARM) measures the strength of an ERP in response to an auditory stimulus. The calculation of eARM is a modified version of the ARM calculation (detailed above) that accommodates for the nature of the ERP waveform: MUA ARM calculates ARMs and then averages them across trials, whereas eARM is calculated after averaging ERP

waveforms across trials (to derive a stable ERP signal). eARM is calculated by using the processed waveforms obtained by averaging the first 25 stimulus presentations (for each stimulus separately); averaging the first 25 presentations resulted in a sufficiently stable ERP that was not further affected by adaptation dynamics imposed by stimulus repetitions. To determine the relative difference in response strength between hemispheres for a given bird at a given time-bin, a Lateralization Index (LI) was calculated as the quotient of the absolute difference between each hemisphere's averaged eARMs and their arithmetic mean (Eq. 1). Additionally, to compare LIs between groups and across birds, baseline LIs were averaged and subtracted from each subsequent LI value; this was done for each bird by using their own baseline activity. This correction was made to normalize the data such that each bird's LI, throughout passive exposure, was controlled by their own levels of activity during baseline.

$$LI = \frac{R - L}{[\frac{R+L}{2}]} \tag{1}$$

**Operant Go-NoGO training procedure**. After the termination of the period of passive exposure, subjects were removed from their isolation boxes and placed in an operant training chamber (Supplementary Fig. 1). Subjects were acclimated to their new boxes and shaped to peck a sensor. All shaping and conditioning contingencies were controlled using ARTSy[70]. Once subjects were pecking reliably (1–2days), training on the Go/NoGo paradigm commenced. Discrimination stimuli were two novel canary songs (duration = 1.2 s, 70 dB) that were 70% similar to each other (%similarity measure in Sound Analysis Pro[71]). One song was arbitrarily assigned as the Go stimulus, and another song was assigned as the NoGo stimulus (counterbalanced across subjects).

The apparatus contained an infrared beam (the response key) and a retractable food hopper. Subjects had to break the beam once to initiate a trial; the approach broke the infrared beam and a response was recorded. Once a trial was initiated, a Go or NoGo stimulus (50/50 probability per 50 trial-blocks) was played through the speakers (Supplementary Fig. 1). If the Go song was presented, subjects had 4 s to respond with a second beam-break in order to obtain access to a food reward from the extended food hopper for 20 s. If the NoGo song was played, subjects had to withhold responding to avoid receiving a 10 s lights-out timeout punishment. All trials terminated after 6 s if there was no response; in a NoGo trial, not responding within this time was considered a correct behavior. While in the operant training chambers, subjects had access to the training apparatus for 8 h a day and received 4 h of free access to food during the non-training period. Subjects were housed in the operant training chamber for 5 days, then they were returned for 2 days to their passive environment box where they received refresher playback of their respective auditory environments (Con-Env or Het-Env). Training continued until subjects reached 80% criterion on 2 consecutive blocks of 50 trials each or until 21 days of operant training had elapsed, whichever came first and regardless of final performance scores. Final performance on the operant discrimination test was defined as the percentage of correct responses on the day subjects reached or exceeded 80% criterion; the number of days to reach criterion was recorded.

**Statistics and reproducibility**. All statistical analyses were performed using OriginPro (2020b) or Statistica (2015). The time-course of the transient shifts in lateralization was assessed by using the LI measure and by compartmentalizing time into 3-day bins. The nature of the eARM LI set of data did not allow for a repeated measures analytical approach due to an unequal number of data points per time bin. Therefore, the eARM LI data was analyzed in two approaches: factorial ANOVA and *post-hoc* comparisons. To compare time, acoustic environment, and their interactive effect on lateralized patterns of activity, a $2 \times 8$ ANOVA was employed with time bins and exposure type as fixed factors. *Post hoc* comparisons, via independent t-tests, between Con-Env and Het-Env at discrete time bins were carried out following the omnibus test. One bird from the Het-Env group of the eARM LI data set was dropped from the statistical analysis due to highly variable and inconsistent data.

Main and interactions effects of auditory environment (Con-Env, Het-Env) and hemisphere (left and right) on MUA-derived data were analyzed by using repeated measures ANCOVA on ARM and adaptation rate measures with stimulus category (zfstim, canstim) as the repeated measure. ARMs tend to decrease with depth along the dorsal-ventral axis; therefore, our analyses included depth as a covariate when applicable.

Behavioral performance in the auditory discrimination task was compared between Con-Env and Het-Env groups that were exposed to their respective environment for different duration. The main effects of, and interactions between, auditory environment and duration of exposure were compared via ANOVAs with final percentage scores, the performance score on the final day of training, as the dependent measures.

Analyses on the potential effect of ground wire on lateralized epidural activity were carried out by analyzing baseline rms between hemispheres or groups via non-parametric Kruskal-Wallis H-test.

The criterion of significance for all omnibus tests was set to .05 and the criterion for *post hoc* comparisons was Bonferroni-corrected for the appropriate number of comparison-pairs. All two-group or *post-hoc* comparisons were performed as two-sided independent-samples t-tests.

All custom scripts employed to analyze the electrophysiology data can be found in a Zenodo repository[72].

**Reporting summary**. Further information on research design is available in the Nature Portfolio Reporting Summary linked to this article.

## Data availability

Datasets used to generate graphical displays and conduct statistical analyses are available upon request to the corresponding author. The source data behind the graphs in the paper can be found in Supplementary Data 1.

## Code availability

Any custom scripts employed in the analyses of the datasets are available upon request and/or are available in a Zenodo repository[72].

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

## Acknowledgements

D.V. and B.F.C. acknowledge Drs. Kasia Bieszczad, John McGann, Kai Lu, and Mimi Phan for thoughtful comments and feedback on earlier versions of the manuscript. This work was supported by National Institute on Deafness and Other Communication Disorders (DC008854).

## Author contributions

L.Y., D.V., and B.F.C. designed different components of the research. L.Y. performed a set of acute electrophysiology experiments and B.F.C. completed the chronic experiment as well as auxiliary acute electrophysiology experiments. J.O. and B.C. helped with the acquisition of the chronic electrophysiology data. Data analysis was performed by B.F.C. and L.Y. with input from D.V.; B.F.C., L.Y., and D.V. wrote the paper.

## Competing interests

The authors declare no competing interests.
