## [Peer Review File · Communications Biology]

Reviewers' comments:

Reviewer #1 (Remarks to the Author):

This study substantively expands on previous work characterizing experience-dependent changes in lateralized auditory processing in the zebra finch forebrain. It examines the effects of prolonged (14-30 day) ambient exposure to a novel acoustic environment (canary aviary sound). The authors show that after an initial, previously-reported, reversal from right- to left-lateralized response bias, extended exposure to the novel acoustic environment results in a reversion back to the typical right-biased pattern. This is measured using two methods: 1) longitudinal epidural recordings of ERPs in response to female zebra finch calls, across 20 days of passive exposure to either canary or zebra finch aviary sounds, and 2) acute bilateral multiunit recordings in NCM during the presentation of zebra finch songs or canary songs for cohorts of birds exposed to either 4,9,14, or 30 days of conspecific or heterospecific acoustic environments. For both methods, a lateralization index was computed based on playback-evoked response magnitudes, and for MUA in NCM, stimulus adaptation rates were calculated. Furthermore, the authors show that the typical pattern of right-biased activity in NCM is abolished if birds are deprived of any complex environmental sounds, but that the bias returns after resumed exposure.

The authors also present intriguing behavioral data from separate cohorts of birds, in which those that received prolonged exposure to conspecific sounds are better able to discriminate between conspecific songs in a go-nogo task. This is a nice demonstration that passive exposure to novel sensory statistics can affect behavioral performance. Although not directly tested, this behavioral benefit is presumed to correspond to the changes in lateralized neural responses observed in the ephys cohorts.

Overall, the study provides important new insights into experience-dependent changes in sensory processing and behavior in adulthood. Because zebra finches serve as a useful model for studying multiple aspects of vocal communication, with analogies to speech production and perception in humans (including lateralization), these results may be of general interest to the broader readership of communications biology. The manuscript is well-written, with thoughtful interpretations and discussion. The data are, for the most part, clearly presented, however a few important points should be addressed.

1) There are multiple instances in which relevant data or analyses are described but not shown.

a) In the methods section it is stated that stimulus specific adaptation is observed in the caudal-most ERPs (LL 607-609) but adaptation rates are not presented or analyzed for the ERPs/longitudinal recordings. Is it possible, for example, that the trend toward lateralization across caudal sites might be more clearly observed in exposure-dependent changes in adaptation rates? Similarly, I'm curious if it wouldn't be possible to determine the direction of ADR change across hemispheres (LL 331-333) based on ERP adaptation rates.

b) Data for MUA responses to various playbacks types should be shown. Multiple differences in responses across stimulus types are described (e.g. LL 234-267, 344-345) but not displayed in any figure. Experience-dependent changes to HET responses or differences from/in CON responses (eg. LL 242-243) are relevant for interpreting the effects of exposure. Visually unpacking these results would bring some much-needed clarity to an otherwise dense section of analyses. Similarly, it would be informative to show the average MUA responses/tuning distributions derived from the pure-tone playbacks (LL 648-658): i) are there hemispheric biases for certain frequencies? ii) does tuning change with HETENV exposure? iii) how does tuning relate to spectral characteristics of HET/CON stimuli and their responses? This might provide low feature-level evidence for the flexible perceptual filters described in the intro and discussion.

2) In general, greater symmetry in the presentation of the MUA and ERP data would facilitate

comparisons and potentially reveal complementarity of the methods (in addition to establishing consistency of results). Along these lines, it should be noted that for MUA, HET and CON song playbacks were used but for ERPs, only CON female call playbacks were presented. The rationale for/limitations of assessing effects of HETENV using only on CON stimuli should be discussed, in particular when attempting to associate changes in CON responses to the facilitation of HET-related behavior.

3) For the ERP data, I am a bit concerned that the left-lateralized position of the ref/ground wire (LL 572-573 & Fig 9) might have a systematic influence on the Absolute Response Magnitudes across sites (e.g. ERPs on channels closer to reference tending to have lower amplitudes):

"The practical importance of referencing for ERP users is that the voltage measured at active electrode sites closer to the reference site will necessarily be closer to 0 V, all other things being equal." From A brief introduction to the use of event-related potentials in studies of perception and attention, G.F. Woodman, 2010

Based on the description of the eARM measure in the methods (LL 696-697) "The calculation of eARM is a modified version of the ARM calculation (detailed above) that accommodates for the nature of the ERP waveform" it is not clear if this measure, and subsequent lateralization index, could be biased by the use of a single asymmetric reference.

Possible ways to address concerns:

- a) Please provide additional clarity regarding how the ARM measure is specifically modified to accommodate ERP waveforms and why this would not be susceptible to reference-related differences in ERP amplitude.
- b) Provide additional detail/examples for the raw epidural signals and their baseline variability across channels, examining the effects of off-line re-referencing if possible.
- c) Include relevant citations of previous eARM comparisons across epidural recording sites.
- d) Include additional direct comparisons of MUA and ERP data (e.g. of LI ranges and variability).

4) The proposed interpretation that the return to typical lateralization reflects memory consolidation for new, but now familiar stimulus statistics is intriguing. I would be interested in knowing more about how the authors envision this process based on known structural and functional connectivity patterns across hemispheres (also addressing the fact that zebra finches lack a corpus callosum).

5) Figures 1 and 9 contain illustrative elements that are in some cases identical to those in the group's previous work (Yang & Vicario, 2015, Neuroscience) and should at least be cited as such (publisher's permission to reproduce is likely required). This includes the presentation of the same example MUA traces (Fig. 9D). Although the authors state that MUA data collected at 4d and 9d were previously published, it is not immediately evident where these were combined with new data.

Minor Comments:

Fig 6:

- Color key/descriptions in figure legend are missing.
- Display the population variability for each time point. Also it is unclear which time points represent the same population of birds: "...hashed lines denote independent groups, within the same group, recorded at discrete durations of exposure." This wording is confusing. For each exposure type (HETENV or CONENV), independent groups were recorded for each of the 4 exposure periods?

Fig 7:

- Legend refers to top and bottom panels but coincides with subplots A and B which are presented side-by-side.

- Based on legend text, lines indicated "30d" in the key include 14d as well and should be indicated as such.
- Consistent use of abbreviations would be helpful (i.e. "9d HETENV/CONENV" in the key instead of "9d can/zf").

Figure 8 is missing entirely but callouts to Fig 8 in the methods seem to be referring to elements of Fig 1.

Presenting related data as subplots of a single figure for (e.g. 2 & 4 & 5) would enable easier comparison.

Figure 9:

- 9A has an incomplete label: "Inner and Outer Layers of".
- 9C Axis tick labels are illegible.
- 9D Time and voltage scale bars are missing.

Reviewer #2 (Remarks to the Author):

This study in male zebra finches shows neural plasticity for lateralization when listening to the song of conspecifics. The NCM is shown by earlier studies from the same laboratory to be lateralized for processing complex vocalization, in this case of calls made by conspecifics. The data obtained using both epidural recording of neural activity as well as multiunit recordings using depth electrodes suggests that lateralization is transiently reversed when birds listen to or are exposed to the sound of a novel heterospecific (canary) vocalizations for a period of ~ 21 days. This happens during a time window of four to nine days postexposure. After about two weeks, lateralization once again returns to its original status. The data suggest that exposure to novel sounds primes the NCM on the left to start responding strongly to conspecific vocalizations. The extended exposure of 14 days or more also enhances auditory discrimination for heterospecific stimuli. Another interesting result is that continuous exposure to complex sounds is needed for the typical lateralized patterns observed during the silent condition to be maintained over time.

The results of this study are original and very interesting as they show that lateralization is not a static phenomenon, rather there is a shift in lateralization during and soon after exposure to the vocalization of conspecifics. This is quite intriguing and questions the functionality of lateralization as a static processing scheme. The figures are appropriately formatted and organized. Some rewriting of the discussion is highly recommended, however, to improve the scholarship and scientific impact of the study.

The authors explain their findings by comparing their results to those of language learning in humans. While the results are interesting and experimental data and statistical analyses are fine, their interpretation is not handled well. There are only few examples showing lateralization of neural responses to communication sound processing in nonhuman species. Rather than trying to explain results of this study in a songbird to language learning in humans, it is more appropriate to compare lateralization observed within neural activity first and foremost to that in nonhuman species. This omission suggests a lack of scholarship and an overreach in expressing the significance of the findings in the context of human language, notwithstanding previous comparisons made by some between bird song and language that may be valid within certain contexts. The findings may apply as well for speech sound processing, but less so for language per se in humans. As has been shown, the lateralization in both birds, bats and nonhuman primates is driven by ethological and functional

constraints and acoustics (Belin et al., 1998; Cousillas et al., 2020; Kanwal, 2012; Poremba et al., 2004; Rogers et al., 2004). In this context, it could apply equally well or better to musical sounds and more importantly prosody in speech (Christensen-dalsgaard, 2004; Stiller et al., 1997). Neurophysiological studies in both bats and humans show that a difference in the lateralization is based upon the time windows for processing sound (perceptual filters) (Belin et al., 1998; Kanwal, 2012; Poeppel, 2003; Washington & Kanwal, 2012), namely a bias for longer time windows within which to process sounds in the right hemisphere and shorter time windows in the left hemisphere. This common computational constraint in both bats and humans translates into lateralization for processing echolocation vs. communication sounds in bats and music vs. speech in humans.

At a relatively young age, humans, bats and birds exhibit babbling (Fernandez et al., 2021; Ter Haar et al., 2021), which is more closely related to the production and perception of prosody and eventually translates into the ability to recognize and produce species-specific communication sounds (Saksida et al., 2021). Thus, one hemisphere, the right in mammals and presumably the left in birds (this study) may become more active after exposure to heterospecific sound to allow parsing a sound stream before analyzing it at higher resolution in the time domain. The results of this study appear to be consistent with this interpretation but a discussion of this literature is entirely missing. Including this will greatly improve the relevance and impact of the study. The data suggest that a hemispheric difference in temporal windows for processing complex sounds is both dynamic and plastic, being maintained by the exposure to conspecific sounds but altered by continuous exposure to heterospecific sounds. In this way, hemispheric asymmetry may not just be limited to developmental stages, but also extend to the learning of new sounds in adults. This may have important implications also for recovery from post-stroke aphasia, suggesting the possibility of reactivating the remaining tissue possibly by activating the right (nondominant) auditory cortex in young adults (Newport et al., 2022).

In short, the study is well conceived and conducted, but the interpretation of the data with an orientation towards language rather than acoustics and related neural mechanisms is misleading for future research directions and lessens its impact.

Abstract: Change "As for human language, auditory processing in songbirds' ... " to "As for speech sound processing in humans, auditory processing in songbirds ..."

CONENV and HETENV and related acronyms will be easier to follow if formatted as Con-Env and Het-Env since each letter does not stand for a different word. Similarly, AV-SIL-S is better abbreviated as Av-Sil-S.

Line 531: Was each bird placed in acoustic isolation from others?

In Figure 1A, please also show the amplitude envelop of each sound stream. To what extent is the hemispheric shift affected by sound amplitude?

Literature references:

- Belin, P., Zilbovicius, M., Crozier, S., Thivard, L., Fontaine, A., Masure, M. C., & Samson, Y. (1998). Lateralization of speech and auditory temporal processing. *Journal of Cognitive Neuroscience*, 10(4), 536–540. <https://doi.org/10.1162/089892998562834>
- Christensen-dalsgaard, J. (2004). Jakob Christensen-Dalsgaard Music and the Origin of Speeches. *Journal of Music and Meaning*, 2.
- Cousillas, H., Henry, L., George, I., Marchesseau, S., & Hausberger, M. (2020). Lateralization of social signal brain processing correlates with the degree of social integration in a songbird. *Scientific Reports*, 10(1), 14093. <https://doi.org/10.1038/s41598-020-70946-7>

Fernandez, A. A., Burchardt, L. S., Nagy, M., & Knörnschild, M. (2021). Babbling in a vocal learning bat resembles human infant babbling. *Science*, 373(6557), 923–926. <https://doi.org/10.1126/science.abf9279>

Kanwal, J. S. (2012). Right-left asymmetry in the cortical processing of sounds for social communication vs. navigation in mustached bats. *The European Journal of Neuroscience*, 35(2), 257–270. <https://doi.org/10.1111/j.1460-9568.2011.07951.x>

Newport, E. L., Seydell-Greenwald, A., Landau, B., Turkeltaub, P. E., Chambers, C. E., Martin, K. C., Rennert, R., Giannetti, M., Dromerick, A. W., Ichord, R. N., Carpenter, J. L., Berl, M. M., & Gaillard, W. D. (2022). Language and developmental plasticity after perinatal stroke. *Proceedings of the National Academy of Sciences of the United States of America*, 119(42), e2207293119. <https://doi.org/10.1073/pnas.2207293119>

Poeppel, D. (2003). The analysis of speech in different temporal integration windows: cerebral lateralization as 'asymmetric sampling in time.' *Speech Communication*, 41(1), 245–255. [https://doi.org/10.1016/S0167-6393\(02\)00107-3](https://doi.org/10.1016/S0167-6393(02)00107-3)

Poremba, A., Malloy, M., Saunders, R. C., Carson, R. E., Herscovitch, P., & Mishkin, M. (2004). Species-specific calls evoke asymmetric activity in the monkey's temporal poles. *Nature*, 427(6973), 448–451. <https://doi.org/10.1038/nature02268>

Rogers, L. J., Zucca, P., & Vallortigara, G. (2004). Advantages of having a lateralized brain. *Proceedings. Biological Sciences / the Royal Society*, 271 Suppl 6(Suppl 6), S420-2. <https://doi.org/10.1098/rsbl.2004.0200>

Saksida, A., Flo, A., Guedes, B., Nespore, M., & Peña Garay, M. (2021). Prosody facilitates learning the word order in a new language. *Cognition*, 213, 104686. <https://doi.org/10.1016/j.cognition.2021.104686>

Stiller, D., Gaschler-Markefski, B., Baumgart, F., Schindler, F., Tempelmann, C., Heinze, H. J., & Scheich, H. (1997). Lateralized processing of speech prosodies in the temporal cortex: a 3-T functional magnetic resonance imaging study. *Magma*, 5(4), 275–284.

Ter Haar, S. M., Fernandez, A. A., Gratier, M., Knörnschild, M., Levelt, C., Moore, R. K., Vellema, M., Wang, X., & Oller, D. K. (2021). Cross-species parallels in babbling: animals and algorithms. *Philosophical Transactions of the Royal Society of London. Series B, Biological Sciences*, 376(1836), 20200239. <https://doi.org/10.1098/rstb.2020.0239>

Washington, S. D., & Kanwal, J. S. (2012). Sex-dependent hemispheric asymmetries for processing frequency-modulated sounds in the primary auditory cortex of the mustached bat. *Journal of Neurophysiology*, 108(6), 1548–1566. <https://doi.org/10.1152/jn.00952.2011>.

Response to Reviewers: Authors' comments and responses are written in Times New Roman and highlighted in yellow. Any edits or new additions to the manuscript have been highlighted yellow.

*Line and Figure Numbers have changed following modifications to the manuscript. The lines and figure numbers referenced in the author's reply reflect the new line and figure numbers.

*Note that, in response to Reviewer 2, we have changed the acronyms to a combination of upper and lower case here and in the main text, e.g. HETENV becomes Het-Env, etc...

Dear Reviewers,

The authors profoundly appreciated all the provided comments as they stimulated both new analytical routes and discussions of different perspectives with which to interpret the results. Below are the author's replies to each of the comments, with descriptions or responses where appropriate.

Reviewer 1 Comments,. Referee #1: Birds, electrophysiology, vocal communication

This study substantively expands on previous work characterizing experience-dependent changes in lateralized auditory processing in the zebra finch forebrain. It examines the effects of prolonged (14-30 day) ambient exposure to a novel acoustic environment (canary aviary sound). The authors show that after an initial, previously-reported, reversal from right- to left-lateralized response bias, extended exposure to the novel acoustic environment results in a reversion back to the typical right-biased pattern. This is measured using two methods: 1) longitudinal epidural recordings of ERPs in response to female zebra finch calls, across 20 days of passive exposure to either canary or zebra finch aviary sounds, and 2) acute bilateral multiunit recordings in NCM during the presentation of zebra finch songs or canary songs for cohorts of birds exposed to either 4,9,14, or 30 days of conspecific or heterospecific acoustic environments. For both methods, a lateralization index was computed based on playback-evoked response magnitudes, and for MUA in NCM, stimulus adaptation rates were calculated. Furthermore, the authors show that the typical pattern of right-biased activity in NCM is abolished if birds are deprived of any complex environmental sounds, but that the bias returns after resumed exposure.

The authors also present intriguing behavioral data from separate cohorts of birds, in which those that received prolonged exposure to conspecific sounds are better able to discriminate between conspecific songs in a go-nogo task. This is a nice demonstration that passive exposure to novel sensory statistics can affect behavioral performance. Although not directly tested, this behavioral benefit is presumed to correspond to the changes in lateralized neural responses observed in the ephys cohorts.

Overall, the study provides important new insights into experience-dependent changes in sensory processing and behavior in adulthood. Because zebra finches serve as a useful model for studying multiple aspects of vocal communication, with analogies to speech production and perception in humans (including lateralization), these results may be of general interest to the broader readership of communications biology. The manuscript is well-written, with thoughtful interpretations and discussion. The data are, for the most part, clearly presented, however a few important points should be addressed.

1) There are multiple instances in which relevant data or analyses are described but not shown. a) In the methods section it is stated that stimulus specific adaptation is observed in the caudal-most ERPs (LL 607-609) but adaptation rates are not presented or analyzed for the ERPs/longitudinal recordings. Is it possible, for example, that the trend toward lateralization across caudal sites might be more clearly observed in exposure-dependent changes in adaptation rates? Similarly, I'm curious if it wouldn't be possible to determine the direction of ADR change across hemispheres (LL 331-333) based on ERP adaptation rates.

The authors are also interested in adaptation of ERP responses which, together with the Reviewer's comments, sparked new analyses. However, after extensive reanalysis, the ERP-adaptation related data lack consistency and/or were not sufficiently convincing or comparable to the MUA AdR data to justify inclusion. First, the initial analysis—which led to mentioning (in the original version of the manuscript) that caudal-pins demonstrate adaptation—was based on averaging over 25 repetitions of each stimulus (out of 100) of a given stimulus, leading to 4 chunks of eARM data); this was meant to be a qualitative observation rather than a precise analysis, given that we were averaging across long repetition bouts, over which responses are known to adapt at the neuronal level (Chew et al., 1995). Therefore, any adaptation analysis for the ERP data, via this method, would not be comparable to MUA AdRs. Second, per the Reviewer's suggestion, we attempted to quantify and compare ERP adaptation and MUA AdR over fewer repetitions (averaging ERPs over ~5 repetitions of each stimulus) in order to maintain a reasonable correspondence between ERP and MUA adaptation rates. Adaptation of ERPs over successive chunks was quantified in three different ways: 1) the raw slope in root-mean-square (rms) as a function of chunk, 2) the slope corrected by the average rms across all chunks, and 3) the slope corrected by the average absolute response magnitude (arm) across chunks; methods 2 and 3 are more closely related to the MUA AdR computation. Finally, these metrics were plotted for each bird and group as a function of recording session (during Het-Env or Con-Env exposure). However, while some of these analyses suggested a reduction in response amplitude consistent with adaptation, the results were not sufficiently convincing to provide clear evidence of adaptation that could be compared to the MUA data. Thus, the intrinsic variability in ERP waveforms is a serious limiting factor. In addition, the lack of consistent adaptation seen in the ERP reanalysis may be due to the fact that activity on the caudal epidural pins originates in the auditory lobule, of which NCM is only a part. The next most rostral area, Field L, does not show adaptation. Given the interpretational difficulties, and anatomical (non-NCM sources) influences on the measurement of responses, we propose to leave out the ERP adaptation analyses from the main manuscript.

b) Data for MUA responses to various playbacks types should be shown. Multiple differences in responses across stimulus types are described (e.g. LL 234-267, 344-345) but not displayed in any figure. Experience-dependent changes to HET responses or differences from/in CON responses (eg. LL 242-243) are relevant for interpreting the effects of exposure. Visually unpacking these results would bring some much-needed clarity to an otherwise dense section of analyses. Similarly, it would be informative to show the average MUA responses/tuning distributions derived from the pure-tone playbacks (LL 648-658): i) are there hemispheric biases

for certain frequencies? ii) does tuning change with HETENV exposure? iii) how does tuning relate to spectral characteristics of HET/CON stimuli and their responses? This might provide low feature-level evidence for the flexible perceptual filters described in the intro and discussion.

Per suggestion, the MUA data was reanalyzed (LL 273-284) and a graphical display (New Figure 5) was produced in order to demonstrate that the shifts in lateralized activity occur regardless of the test stimulus (canary or ZF songs). As seen in the graph, indeed when lateralization changes in 4/9D Het-Env-exposed birds (left panel), it is observed for either test stimulus, be it canary or ZF songs. Nevertheless, ZF-elicited always seems to be higher (when hemisphere factor is unaccounted for), suggesting conspecific bias for ZF song, a previously observed phenomenon in the ZF.

There is a suggestion that frequency tuning distributions shift during Het-Env playback (Yang & Vicario, 2015), however any differences should be interpreted lightly: 1) sampling biases may influence tuning distributions, and 2) we could not assume that we were recording from sufficiently similar neuron distributions (ergodicity issue). While we share the reviewer's suggestion, such an analysis would require a higher sample size to determine whether frequency tuning distributions reliably and consistently shift as a function of Het-Env exposure by either sampling from sufficiently similar neuron distributions and/or in a chronic assessment of NCM MUA in a similar experiment. Certainly, we suspect that tuning distributions (as has been observed in a similar study of the lab; Terleph, Lu, & Vicario, 2008) may be shifting as a function of environmental exposure and plan on conducting a separate set of experiments by chronically sampling NCM MUA in the same birds exposed to different acoustic environments (e.g. Het-Env).

2) In general, greater symmetry in the presentation of the MUA and ERP data would facilitate comparisons and potentially reveal complementarity of the methods (in addition to establishing consistency of results). Along these lines, it should be noted that for MUA, HET and CON song playbacks were used but for ERPs, only CON female call playbacks were presented. The rationale for/limitations of assessing effects of HETENV using only on CON stimuli should be discussed, in particular when attempting to associate changes in CON responses to the facilitation of HET-related behavior.

In the manuscript, symmetry in the presentation of results has been enhanced by producing two figures: Figures 2 and 6, both of which plot the data in terms of Lateralization Indices (LIs). While it might be possible to combine these graphs in one figure, we think it will interrupt the flow of the results presentation. We attempted an overlay of the LI graphs from Figures 2 and 6, but the resulting figure was too cluttered and therefore that idea was abandoned. However, it should be noted that, while the LI calculations are the same, the magnitude measures are slightly different and the sources (ERP dura and MUA neuronal) are also very different.

In addressing the second comment, highlighted lines (LL 723-729) in the manuscript have been introduced to explain the rationale for using ZF female calls to elicit ERPs. In short, during an experimental pilot, ZF song was previously used to elicit ERPs. While song stimuli indeed elicited an ERP waveform, the waveform itself is long and difficult to interpret as multiple peaks and troughs are observed, perhaps due to changing stimulus qualities or brief 'silences' that are inherent in ZF songs. Female calls (FCs), as observed in (new) Figure 8, elicit stereotyped ERPs, and the stimulus onset and offset are discrete. In addition, FCs have been previously shown to elicit high responsivity in male ZFs (Vicario, Naqvi, & Raksin, 2001) and, more importantly, are able to drive lateralized responses (Phan & Vicario, 2010) yet are not as complex as song stimuli; since chronically assayed birds (ERP data) were continuously exposed to either Con-Env and Het-Env, which included songs, songs were not used in testing in order to limit possible interaction with the effects of the acoustic environment on lateralized

activity. Lastly, it is relevant to note that even at the level of NCM, both canary and zebra finch test stimuli elicit lateralized responses in the same direction (right- or left-lateralized) as a function of environment exposure; suggesting a more striking set of results, given that conspecific songs are asymmetrically processed as a function of auditory environment and not in a stimulus-specific manner.

3) For the ERP data, I am a bit concerned that the left-lateralized position of the ref/ground wire (LL 572-573 & Fig 9) might have a systematic influence on the Absolute Response Magnitudes across sites (e.g. ERPs on channels closer to reference tending to have lower amplitudes): “The practical importance of referencing for ERP users is that the voltage measured at active electrode sites closer to the reference site will necessarily be closer to 0 V, all other things being equal.” From A brief introduction to the use of event-related potentials in studies of perception and attention, G.F. Woodman, 2010
Based on the description of the eARM measure in the methods (LL 696-697) “The calculation of eARM is a modified version of the ARM calculation (detailed above) that accommodates for the nature of the ERP waveform” it is not clear if this measure, and subsequent lateralization index, could be biased by the use of a single asymmetric reference.

The authors appreciate the reviewer’s comment on how the reference wire may affect the overall amount of activity recorded by the epidural electrode pins. However, three points may suggest that there were no systematic influences of the ground wire. First, the ground wire was placed atop the left hemisphere, far from the epidural pins, for all of the subjects; therefore, given that positioning, the prediction would be that left-hemisphere activity would always be lower than the right, which is not the case. Second, **Figure 2** illustrates that Het-Env-exposed ZFs display shifts in lateralized activity, whereby higher activity in the left hemisphere was observed; no such pattern was observed in Con-Env-exposed birds. This would suggest that it is possible to record higher activity in the left hemisphere even when the ground is positioned on the same side as the left-hemisphere epidural pins. Third, eARMs were replotted for each environment (**new Figure 9**) to determine the distributions of eARMs between hemispheres and the rostrocaudal axis. The Con-Env plot (left-panel) is not surprising, the caudal pins recorded higher activity in the right hemisphere as is typical of MUA activity (suggested by LI; Figure 2). Importantly, the Het-Env plot (right-panel) showed no lateral differences within the caudal pins. This is due to time being collapsed in the plot which then combines the time points when Het-Env induces a left-lateralization with other points that are right-lateralized. Together, we believe that the position of the wire had a negligible effect on the relative amplitude of recorded ERPs and no material effect on the results we report. After all, the wire was always on the left, yet we observed both left and right biased activity at different time points in the same bird.

Possible ways to address concerns:

a) Please provide additional clarity regarding how the ARM measure is specifically modified to accommodate ERP waveforms and why this would not be susceptible to reference-related differences in ERP amplitude.

The wording in the manuscript, “modified to accommodate ERP waveforms”, has been changed along with a description of the difference; the modifications are now in **LL 792-794**. In short, the referred modification is not directly related nor susceptible to reference-related differences. ARMs for MUA activity are calculated for **each** trial, by taking the root mean square of the multi-unit waveform over a sampling period (e.g. stimulus-playback). Then MUA ARM is calculated by taking an average of the rms from trials 2-6 and corrected for average baseline activity (Phan & Vicario, 2010). For eARM, the waveforms were **first** averaged over the **first** 25 presentations of a given stimulus, and **then** the rms was calculated over the stimulus onset period. eARM was then computed as the difference between the stimulus-period rms and the baseline rms. The modification is thus based on when the data is averaged, in

the case of the ERP eARM the waveforms are averaged then rms is calculated, in the case of MUA ARM rms is calculated first and then averaged.

b) Provide additional detail/examples for the raw epidural signals and their baseline variability across channels, examining the effects of off-line re-referencing if possible.

The new Figure 10 includes 4 raw epidural traces of activity, including baseline (period prior to stimulus onset), activity changes during stimulus playback, and return to baseline. To address the reviewer's concern regarding left-lateralized positioning of the reference wire, two histograms were produced to assay the distributions of baseline RMS (baseline RMS is used in eARM calculation to derive the relative amplitude of the signal change in response to stimuli). The first histogram (left panel) illustrates that there are no significant differences between left and right baseline RMS values; there are also no significant differences in baseline RMS when the data is divided between exposure groups (right panel). Both comparisons were conducted via a Kruskal-Wallis H-test to determine differences in distributions (LL 683-701). Together, we interpret these results as the lack of a systematic effect of the left-lateralized reference wire, either between hemispheres (left panel) or exposure groups (right panel). Therefore, the observed eARM LI differences between Con-Env and Het-Env groups are more likely explained by auditory exposure. We appreciate the Reviewer's comment and motivation into our investigation of the possible ground wire effect.

c) Include relevant citations of previous eARM comparisons across epidural recording sites.

To our knowledge, there are no published reports on the use of eARM, only MUA-derived ARMs in (Phan & Vicario, 2010) which was adapted (described in previous response). The literature is sparse, we are aware of two articles in songbirds that employed ERP analysis (e.g. Maul et al., 2010), yet neither utilized eARM as a measurement nor was hemisphere considered; they were concerned with stimulus effects.

d) Include additional direct comparisons of MUA and ERP data (e.g. of LI ranges and variability).

The author's believe that the raised issue/suggestion (here, 3d) has been addressed in the responses to the reviewer's recommendations in 3, 3a, and 3b. Please advise if the authors should generate such a comparison in the case that the replies to 3, 3a, and 3b are not satisfactory.

4) The proposed interpretation that the return to typical lateralization reflects memory consolidation for new, but now familiar stimulus statistics is intriguing. I would be interested in knowing more about how the authors envision this process based on known structural and functional connectivity patterns across hemispheres (also addressing the fact that zebra finches lack a corpus callosum).

LL 423-440 address this issue in an attempt to tackle the experimental results in the context of songbird brains, which do not possess a corpus callosum. As for the reviewer, the authors have previously been theoretically stimulated by this notion and attempted to reconcile the current results with previously published accounts of interhemispheric communication in the songbird. Specifically, the authors believe that lateralized activity, and changes thereof, may be supported by the anterior commissure, and/or mesencephalic or thalamic gating of ascending information.

5) Figures 1 and 9 contain illustrative elements that are in some cases identical to those in the group's previous work (Yang & Vicario, 2015, Neuroscience) and should at least be cited as such (publisher's permission to reproduce is likely required). This includes the presentation of the same example MUA traces (Fig. 9D). Although the authors state that MUA data collected at 4d and 9d were previously published, it is not immediately evident where these were combined with new data.

We appreciate the reviewer's comments. The authors acquired permissions to utilize the figures and will reference them accordingly. The Results section contains phrasing to the effect that the 4/9D data has been previously published, and the **new Figure 4** (new, concatenated from old figures 4 and 5) caption contains phrasing explaining from where the 4/9D data originates.

Minor Comments:

Fig 6:

-Color key/descriptions in figure legend are missing. **Addressed**
-Display the population variability for each time point. Also it is unclear which time points represent the same population of birds: "...hashed lines denote independent groups, within the same group, recorded at discrete durations of exposure." This wording is confusing. For each exposure type (HETENV or CONENV), independent groups were recorded for each of the 4 exposure periods?

Addressed

Fig 7:

-Legend refers to top and bottom panels but coincides with subplots A and B which are presented side-by-side. **Addressed**
-Based on legend text, lines indicated "30d" in the key include 14d as well and should be indicated as such. **Addressed**
-Consistent use of abbreviations would help (i.e. "9d HETENV/CONENV" in the key instead of "9d can/zf". **Addressed**

Figure 8 is missing entirely but callouts to Fig 8 in the methods seem to be referring to elements of Fig 1. **Apologies, this was a porting error during a pre-submission change of figure order**

Presenting related data as subplots of a single figure for (e.g. 2&10; 4 & 5) would enable easier comparison.

Figures 4 and 5 were initially made separate on purpose but will combine as subplots in the **new Figure 4**. However, the subplots should remain the same as the original graphs (i.e. subplot 1 = 4/9D vs 14D; subplot 2 4/9D vs 30D) because they reflect how the statistical analyses were conducted. We tried combining 4/9D, 14D, and 30D into a single graph but this was not viable since incorporating statistically-significant comparison markers (e.g. asterisks and crossing interaction lines) was not feasible and would be confusing/lead to cluttered graphs.

Figure 9:

-9A has an incomplete label: "Inner and Outer Layers of".

Should say "... of skull", has been addressed

-9C Axis tick labels are illegible.

The figure has been updated to make the labels more legible

-9D Time and voltage scale bars are missing.

Included now, as well as added MUA for canary and ZF stimuli as suggested in prior comment

Reviewer 2 Comments,, Referee #2: mammals, vocal communication, lateralization

This study in male zebra finches shows neural plasticity for lateralization when listening to the song of conspecifics. The NCM is shown by earlier studies from the same laboratory to be lateralized for processing complex vocalization, in this case of calls made by conspecifics. The data obtained using both epidural recording of neural activity as well as multiunit recordings using depth electrodes suggests that lateralization is transiently reversed when birds listen to or are exposed to the sound of a novel heterospecific (canary) vocalizations for a period of ~ 21 days. This happens during a time window of four to nine days postexposure. After about two weeks, lateralization once again returns to its original status. The data suggest that exposure to novel sounds primes the NCM on the left to start responding strongly to conspecific vocalizations.

Thank you for this summary, but it is important to note that heterospecific vocalizations also elicit stronger responding in left NCM at the same time points, and that the return to typical lateralization occurs for both conspecific and heterospecific vocalizations.

The extended exposure of 14 days or more also enhances auditory discrimination for heterospecific stimuli. Another interesting result is that continuous exposure to complex sounds is needed for the typical lateralized patterns observed during the silent condition to be maintained over time.

The results of this study are original and very interesting as they show that lateralization is not a static phenomenon, rather there is a shift in lateralization during and soon after exposure to the vocalization of conspecifics. This is quite intriguing and questions the functionality of lateralization as a static processing scheme. The figures are appropriately formatted and organized.

We thank the Reviewer for recognizing the potential contribution of our results to ideas about the function of lateralized responses.

Some rewriting of the discussion is highly recommended, however, to improve the scholarship and scientific impact of the study.

The authors explain their findings by comparing their results to those of language learning in humans. While the results are interesting and experimental data and statistical analyses are fine, their interpretation is not handled well. There are only few examples showing lateralization of neural responses to communication sound processing in nonhuman species.

We thank the reviewer for pointing out ways in which we can improve our discussion of relevant literature in other nonhuman species. We have now made changes to the Discussion to reflect the Reviewer's suggestions. We have also added text to the Introduction (lines 80-83) to explicitly include the presence of lateralized processing in other non-human species.

Rather than trying to explain results of this study in a songbird to language learning in humans, it is more appropriate to compare lateralization observed within neural activity first and foremost to that in nonhuman species. This omission suggests a lack of scholarship and an overreach in expressing the significance of the findings in the context of human language, notwithstanding previous comparisons made by some between bird song and language that may be valid within certain contexts. The findings may apply as well for speech sound processing, but less so for

language per se in humans. As has been shown, the lateralization in both birds, bats and nonhuman primates is driven by ethological and functional constraints and acoustics (Belin et al., 1998; Cousillas et al., 2020; Kanwal, 2012; Poremba et al., 2004; Rogers et al., 2004). In this context, it could apply equally well or better to musical sounds and more importantly prosody in speech (Christensen-dalsgaard, 2004; Stiller et al., 1997).

We did not presume to claim an analogy with human language processing; any such suggestion was inadvertent; the analogy only applies to speech. Our Discussion of lateralization in the context of second language (L2) learning was not meant to make a language analogy. After all, L2 learning involves recognizing and producing novel speech sounds, as well as learning semantics and meaning. While it is tempting to think about relationships between song and prosody, song and music, or song and babbling, we think there are distinctions. While “prosodic” modification might be added to songs (tempo, repetition, emphasis), we have shown elsewhere that the NCM system recognizes and remembers the sounds of specific individuals, so it is not exclusively prosodic. Young songbirds also go through a “babbling” phase as they learn to control their vocal apparatus, and then use it to imitate progressively the sounds they hear from caregivers. We do think the songbird model bears particular relevance to the way communication signals that are culturally acquired through learning (as is true in humans and only very few other species) are processed in the brain.

In any case, in response to the Reviewer’s concerns, we now cite many of the suggested references and include a discussion of the way that acoustic features of song may interact with hemispheric differences in sensitivity to temporal transitions (lines 463-477).

Neurophysiological studies in both bats and humans show that a difference in the lateralization is based upon the time windows for processing sound (perceptual filters) (Belin et al., 1998; Kanwal, 2012; Poeppel, 2003; Washington & Kanwal, 2012), namely a bias for longer time windows within which to process sounds in the right hemisphere and shorter time windows in the left hemisphere. This common computational constraint in both bats and humans translates into lateralization for processing echolocation vs. communication sounds in bats and music vs. speech in humans.

The Reviewer correctly points out that lateral differences in auditory processing for different scales of temporal modulation have been found in humans, bats, and monkeys (with faster transitions, typical of consonants, on the left). And, indeed, those differences contribute to the way that responses to species-typical signals are lateralized. However, those differences are not absolute, and communication signals (including speech) also carry information over slower times scales of rhythm, emphasis, and speech prosody. We also acknowledge that there is no direct evidence in the songbird for hemispheric differences in processing different scales of temporal modulation that corresponds to what has been described for mammals.

At a relatively young age, humans, bats and birds exhibit babbling (Fernandez et al., 2021; Ter Haar et al., 2021), which is more closely related to the production and perception of prosody and eventually translates into the ability to recognize and produce species-specific communication sounds (Saksida et al., 2021). Thus, one hemisphere, the right in mammals and presumably the left in birds (this study) may become more active after exposure to heterospecific sound to allow parsing a sound stream before analyzing it at higher resolution in the time domain.

We agree with the principle but are hesitant to suggest that the higher activity in the right hemisphere of the ZF in response to song necessarily means that the left hemisphere plays a complementary (but reversed) role. We now address a version of this idea (lines 478-491)

The results of this study appear to be consistent with this interpretation but a discussion of this literature is entirely missing. Including this will greatly improve the relevance and impact of the study. The data suggest that a hemispheric difference in temporal windows for processing complex sounds is both dynamic and plastic, being maintained by the exposure to conspecific sounds but altered by continuous exposure to heterospecific sounds. In this way, hemispheric asymmetry may not just be limited to developmental stages, but also extend to the learning of new sounds in adults.

We agree.

This may have important implications also for recovery from post-stroke aphasia, suggesting the possibility of reactivating the remaining tissue possibly by activating the right (nondominant) auditory cortex in young adults (Newport et al., 2022).

Yes.

In short, the study is well conceived and conducted, but the interpretation of the data with an orientation towards language rather than acoustics and related neural mechanisms is misleading for future research directions and lessens its impact.

Abstract: Change “As for human language, auditory processing in songbirds’ ... “ to “As for speech sound processing in humans, auditory processing in songbirds ...”

This change has been made.

CONENV and HETENV and related acronyms will be easier to follow if formatted as Con-Env and Het-Env since each letter does not stand for a different word. Similarly, AV-SIL-S is better abbreviated as Av-Sil-S.

These changes have been made.

Line 531: Was each bird placed in acoustic isolation from others?

Yes. This was stated (new lines 552-554), but has been re-emphasized (lines 596-597).

The original ms stated “birds were individually housed in sound-proof isolation boxes” (original lines 511-513) but the reviewer seems to have missed it.

In Figure 1A, please also show the amplitude envelop of each sound stream. To what extent is the hemispheric shift affected by sound amplitude?

We now include the amplitude envelope for a section of each sound stream. Average amplitude was equated for the two environments and we did not systematically test different amplitudes.

REVIEWERS' COMMENTS:

Reviewer #1 (Remarks to the Author):

I appreciate the authors' thorough responses to my concerns, which have all been sufficiently addressed. I have no additional major comments and commend the authors on their improved manuscript.

Minor formatting points:

-Figure 10A, x-axis label, presumably "Time (s)", is missing.

-Lines 434-435, as well as most figures labels, use the HETENV/CONENV convention, as opposed to the new Het-Env/Con-Env format suggested by Reviewer 2. Usage should be consistent, either way.

Reviewer #2 (Remarks to the Author):

The authors have seriously considered all reviewer comments, and conducted additional analysis of data and generated new figures as well as updated others. This has greatly improved the manuscript. The findings reported are both original and interesting for a wide audience and are now discussed accordingly.

In a few places and in the figures, the old, all caps acronyms still need to be updated.

Response to Reviewers: Authors' comments and responses are written in Times New Roman and highlighted in yellow. Any edits or new additions to the manuscript have been highlighted yellow.

*Line and Figure Numbers have changed following modifications to the manuscript. The lines and figure numbers referenced in the author's reply reflect the new line and figure numbers.

*Note that, in response to Reviewer 2, we have changed the acronyms to a combination of upper and lower case here and in the main text, e.g. HETENV becomes Het-Env, etc...

Response to 2nd round of reviewer comments

Reviewer 1 Comments,. Referee #1: Birds, electrophysiology, vocal communication

I appreciate the authors' thorough responses to my concerns, which have all been sufficiently addressed. I have no additional major comments and commend the authors on their improved manuscript.

Minor formatting points:

-Figure 10A, x-axis label, presumably "Time (s)", is missing.

-Lines 434-435, as well as most figures labels, use the HETENV/CONENV convention, as opposed to the new Het-Env/Con-Env format suggested by Reviewer 2. Usage should be consistent, either way.

The authors appreciate all of the feedback and comments received by Reviewer 1 on both revision rounds, all of the new changes have improved manuscript.

The minor formatting points have been addressed by updating figure 10A and all references to HETENV and CONENV, in-text and on graphical displays, have been switched to Het-Env and Con-Env, respectively.

Reviewer 2 Comments, Referee #2: mammals, vocal communication, lateralization

The authors have seriously considered all reviewer comments, and conducted additional analysis of data and generated new figures as well as updated others. This has greatly improved the manuscript. The findings reported are both original and interesting for a wide audience and are now discussed accordingly.

In a few places and in the figures, the old, all caps acronyms still need to be updated.

The authors appreciate all of the feedback and comments received by Reviewer 2 on both revision rounds, all of the new changes have improved manuscript.

The minor formatting points have been addressed by updating figure 10A and all references to HETENV and CONENV, in-text and on graphical displays, have been switched to Het-Env and Con-Env, respectively.